# Curvature Diversity-Driven Deformation and Domain Alignment for Point Cloud

**Mengxi Wu**                                                      *mengxiwu@usc.edu*
*Department of Computer Science*
*University of Southern California*

**Hao Huang**                                                       *hh1811@nyu.edu*
*Department of Computer Science*
*New York University*

**Yi Fang**                                                          *yfang@nyu.edu*
*Department of Computer Science*
*New York University*

**Mohammad Rostami**                                              *mrostami@isi.edu*
*Department of Computer Science*
*University of Southern California*

**Reviewed on OpenReview:** *https://openreview.net/forum?id=ePXWnH7rGk&referrer*

## Abstract

Unsupervised Domain Adaptation is crucial for point cloud learning due to geometric variations across different generation methods and sensors. To tackle this challenge, we propose **Curvature Diversity-Driven Nuclear-Norm Wasserstein Domain Alignment (CDND)**. We first introduce a Curvature Diversity-driven Deformation Reconstruction (CurvRec) task, enabling the model to extract salient features from semantically rich regions of a given point cloud. We then propose a theoretical framework for Deformation-based Nuclear-norm Wasserstein Discrepancy (D-NWD), extending the Nuclear-norm Wasserstein Discrepancy to original and deformed samples. Our theoretical analysis demonstrates that D-NWD is effective for any deformation method. Empirical experiment results show that our CDND achieves state-of-the-art performance by a noticeable margin over existing approaches. Our codes are available at: `https://github.com/WMX567/CurvRec_DNWD`.

## 1 Introduction

Adopting deep neural networks (DNNs) for point cloud representation learning has led to significant success across applications such as robotics Maturana & Scherer (2015); Duan et al. (2021), autonomous vehicles Mahjourian et al. (2018); Cui et al. (2021), and scene understanding Zheng et al. (2013); Zhu et al. (2017). However, point clouds captured under different conditions often exhibit substantial variations, leading to performance degradation when DNNs are tested on data from domains different from those used during training. This discrepancy, known as the domain gap, is especially problematic in real-world settings. A straightforward solution—retraining the model on newly labeled data—is typically impractical due to the high cost of manual annotation. Unsupervised domain adaptation (UDA) addresses this issue by transferring knowledge from labeled source domains to unlabeled target domains. While UDA has been well studied in 2D planar domains (e.g., images), its extension to 3D point clouds remains underexplored due to challenges from the irregular, unstructured, and unordered nature of 3D data. These properties amplify geometric discrepancies between source and target domains, making direct adaptation methods less effective.

To overcome these challenges, we propose Curvature Diversity-Driven Nuclear-Norm Wasserstein Domain Alignment (CDND). Our approach is motivated by the observation that not all regions in a point cloud equally contribute to semantic understanding and geometric variation can reflect semantic richness. Key semantic features—such as object part boundaries and edges—often occur in regions with complex geometry. As a result, areas with greater geometric variation tend to carry more meaningful semantic information. Curvature diversity can capture geometric variation by measuring how much curvature changes within a local region. Regions with high curvature diversity—reflecting substantial variation in surface orientation and structure—often correspond to semantically rich components such as edges or jointed structures. In contrast, low-diversity regions are typically flatter and less informative, like planar surfaces or object bases.

Our first contribution is the Curvature Diversity-driven Deformation Reconstruction (CurvRec). We introduce a principled region selection strategy guided by curvature diversity, which we quantify using entropy—a standard statistical measure of uncertainty. Entropy captures how spread out the curvature values are within a region: high entropy indicates diverse curvature and complex geometry, while low entropy suggests geometric uniformity. Leveraging this, CurvRec selectively deforms low-entropy (less informative) regions and preserves high-entropy (semantically rich) ones. This contrasts with prior methods such as Achituve et al. (2021), which apply random deformation, and Zou et al. (2021), which classify fixed high-curvature regions. Our method instead focuses deformation on low-diversity areas, allowing the feature extractor to prioritize learning from semantically rich regions.

Our second contribution is a theoretical framework for the Deformation-based Nuclear-norm Wasserstein Discrepancy (D-NWD). Unlike the conventional NWD, D-NWD incorporates features from both original and deformed samples to align source and target domains. This integration creates a more diverse and robust feature space, improving model generalization under domain shifts. Notably, prior work has not provided a theoretical analysis demonstrating that using the NWD to align features from both original and deformed (or augmented) samples can improve performance on the target domain dataset. Our analysis is the first to demonstrate the effectiveness of this strategy. Our theoretical framework shows that D-NWD effectively reduces the domain gap between source and target domains. Furthermore, our analysis illustrates that D-NWD is generic and adaptable to any deformation method, not just the one presented in this paper. Our theoretical analysis is non-trivial since incorporating features from deformed samples requires us to establish entirely new bounds and prove that the NWD remains effective in this expanded feature space spanning both original and deformed samples.

Extensive experiments on standard classification and segmentation benchmarks demonstrate that CDND achieves state-of-the-art performance compared to existing approaches.

## 2 Related Works

**UDA for Point Clouds** While Unsupervised Domain Adaptation (UDA) has been widely explored for 2D planar image data Ganin & Lempitsky (2015); Mansour et al. (2008); Stan & Rostami (2024a); Du et al. (2024); Kumari & Singh (2024), fewer works address UDA for 3D point clouds and non-planar data, where direct adaptation of 2D techniques proves non-trivial Qin et al. (2019); Achituve et al. (2021); Shen et al. (2022); Zou et al. (2021); Liang et al. (2022); Chen et al. (2023); Katageri et al. (2024); Wei et al. (2024); Wu & Rostami; Fan et al. (2022); Cardace et al. (2023). Qin *et al.* Qin et al. (2019) propose PointDAN, combining local and global domain alignment and introducing the PointDA benchmark for classification. Achituve *et al.* Achituve et al. (2021) present a reconstruction-from-deformation method using PointMixup Chen et al. (2020), and introduce the PointSegDA benchmark for segmentation. Zou *et al.* Zou et al. (2021) design two geometry-based self-supervised tasks for learning domain-invariant features. Fan *et al.* Fan et al. (2022) introduce a reliable pseudo-label voting mechanism along with global-local structure modeling. Shen *et al.* Shen et al. (2022) employ geometry-aware implicit functions for modeling domain-specific variations. Liang *et al.* Liang et al. (2022) introduce a masked local 3D structure prediction task to promote feature invariance via spatial context recovery. Cardace *et al.* Cardace et al. (2023) leverage self-distillation to iteratively refine pseudo labels. Katageri *et al.* Katageri et al. (2024) integrate contrastive learning with optimal transport to improve both alignment and discrimination. Chen *et al.* Chen et al. (2023) develop a boundary point prediction task to enhance robustness via geometric boundary cues. Wei *et al.* Wei et al.

(2024) propose a multi-scale part-based representation capturing both local and global part-level information for better adaptation and generalization. These works reflect a growing trend toward integrating geometric priors with adaptation objectives. Our work differs by proposing a principled deformation strategy rooted in curvature diversity, a sophisticated self-supervised task, and a generalized theoretical framework (D-NWD) applicable to any deformation method, achieving state-of-the-art results.

**Optimal Transport for Domain Adaptation**  Although many works on UDA rely on adversarial learning Tzeng et al. (2017); Tang & Jia (2020); Long et al. (2018); Jian & Rostami (2023) to align the distributions of two domains, many rely on direct distribution matching using metric minimization Wang et al. (2017); Li et al. (2020); Fatras et al. (2021). The Wasserstein distance from optimal transport theory is widely used in domain adaptation due to its ability to capture geometric relationships between distributions Courty et al. (2016; 2017); Redko et al. (2017); Rostami & Galstyan (2023); Nananukul et al. (2024). Gautheron *et al.* Gautheron et al. (2019) propose Wasserstein-guided representation learning and a feature selection method to tackle domain shift. Sliced Wasserstein discrepancy is used in Lee et al. (2019); Stan & Rostami (2024b) to replace the $L_1$ distance in Maximum Classifier Discrepancy Saito et al. (2018), offering more meaningful class-level divergence. DeepJDOT Damodaran et al. (2018) maps source-target pairs via a coupling matrix, while CGDM Du et al. (2021) minimizes cross-domain gradient discrepancies. Xu *et al.* Xu et al. (2020) propose a weighted transport method using spatial prototypes for better sample-level alignment. Fatras *et al.* Fatras et al. (2021) introduce unbalanced optimal transport with mini-batch training for scalability. Our D-NWD builds upon NWD Chen et al. (2022), extending it with deformation-based supervision and theoretical generality.

## 3 Proposed Method

We begin by defining the unsupervised domain adaptation (UDA) problem, followed by an overview of our UDA approach, called Curvature Diversity-Driven Nuclear-Norm Wasserstein Domain Alignment (CDND), in Section 3.1. We then present the details of our main contributions: (1) a Curvature Diversity-based Deformation Reconstruction task, described in Sections 3.2 and 3.3, and (2) a theoretical framework for D-NWD, which extends NWD to incorporate both original and deformed samples, as detailed in Section 3.4 and 4. Note that our theoretical analysis is applicable to any deformation method used.

### 3.1 Problem Formulation

We consider a source domain with labeled samples and a target domain with unlabeled samples which has a different data distribution. Our goal is to develop a UDA method to train a model that accurately predicts labels for the target domain using both the source labeled dataset and the target unlabeled dataset. Let $\mathcal{S}$ represent the source domain, where $X_s^i$ denotes the $i$-th batch of samples and $y_s^i$ their corresponding labels. Similarly, let $\mathcal{T}$ represent the target domain, where $X_t^i$ is the $i$-th batch of samples. The feature space induced by $\mathcal{S}$ and $\mathcal{T}$ is denoted by $\Omega_o$. In addition, we introduce deformed domains $\mathcal{S}^d$ and $\mathcal{T}^d$, with their feature space $\Omega_d$. We assume that $\Omega_o$ and $\Omega_d$ are disjoint subsets of $\mathbb{R}^n$, i.e., $\Omega_o \cap \Omega_d = \emptyset$, with $\Omega_o \cup \Omega_d \subseteq \mathbb{R}^n$. This assumption is generally valid in practice, as the probability of a deformed sample being exactly identical to a original one is negligible. A point cloud from the source domain is denoted as $x_s \in \mathbb{R}^{n \times 3}$ and from the target domain is $x_t \in \mathbb{R}^{n \times 3}$, where $n$ is the number of points. The corresponding deformed point clouds are denoted by $x_s^d$ and $x_t^d$.

The pipeline of our CDND is presented in Figure 1. Our model first uses a feature extractor $E$ to obtain shape features from both source and target point clouds. To minimize domain gaps and ensure domain-invariant features, we: (1) use a curvature diversity-driven deformation reconstruction task via a reconstruction decoder $h_{\text{SSL}}$ and (2) employ the D-NWD to align domains through a classifier $C$. The aligned features are then used for downstream tasks, *i.e.*, point cloud classification and segmentation. The model is trained using source-labeled and target-unlabeled data.

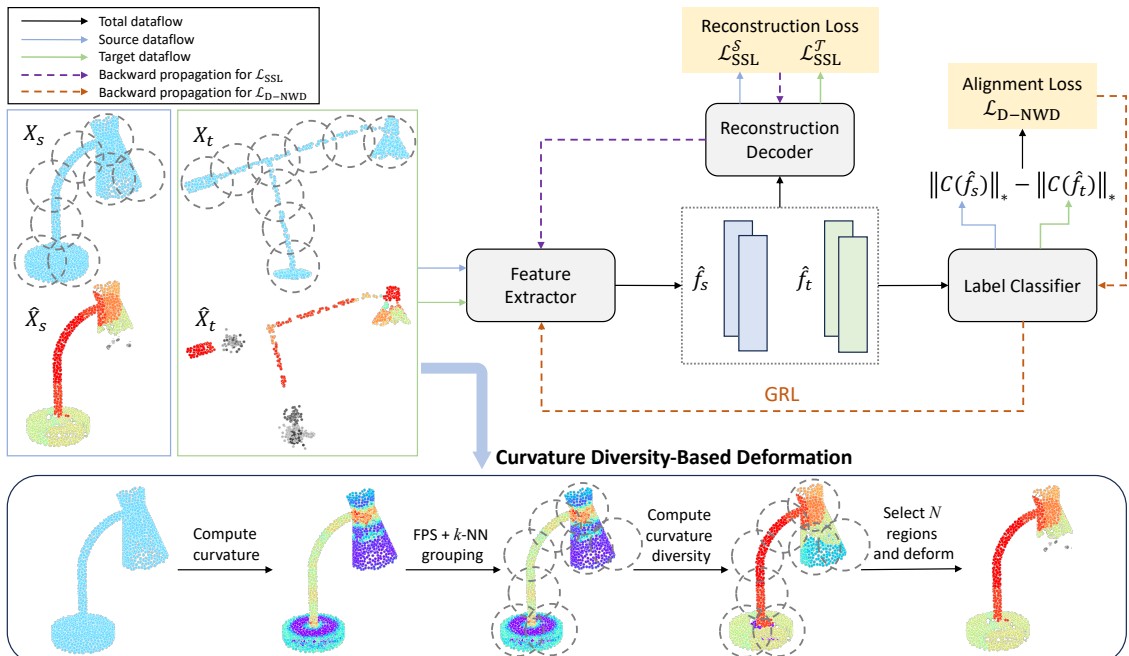

Figure 1: Pipeline of CDND. The inputs are the source batch $X_s$ and target batch $X_t$. We first deform them into $\hat{X}_s$ and $\hat{X}_t$ using Curvature Diversity-Based Deformation. Next, $X_s$, $X_t$, $\hat{X}_s$, and $\hat{X}_t$ are sent into a feature extractor. The features of deformed samples are fed into a reconstruction decoder to reconstruct the deformed regions. For domain alignment, both original and deformed features are sent to D-NWD. Aside from the two losses shown in the figure, a cross-entropy loss is computed on $X_s$ and $\hat{X}_s$ with labels. An NWD loss $\mathcal{L}_{\text{NWD}}^{\mathcal{T}}$ on $X_t$ and $\hat{X}_t$ is also computed to ensure prediction consistency between the target original and deformed pairs.

## 3.2  Curvature Diversity-Driven Deformation

To extract domain-invariant features from point clouds, three deformation strategies have been explored in the literature Achituve et al. (2021): volume-based, feature-based, and sample-based, according to the way of dividing point clouds into regions for deformation. Although these strategies use different techniques to select regions for deformation, they all divide a point cloud into regions based on some spatial locations or arrangements and uniformly randomly select regions to be deformed. However, uniformly random selection may not be optimal as regions within a point cloud vary in their semantic richness, *i.e.*, some regions contain more semantic information that is generalizable across the domains, while some regions may be domain-specific. These semantically rich regions are crucial for tasks such as classification, as they have more distinguishable characteristics. For instance, to differentiate a point cloud of a plant from that of a lamp, focusing on the leaves and flowers — which have richer semantic information — would be more effective than focusing on the flower pot, which is similar to the base of a lamp. Thus, deforming regions with richer semantic information causes the point cloud to lose semantic meaning, making it difficult for a classifier to classify it. Our idea is to identify shared semantic information to improve model generalization in the target domain. To encourage the feature extractor to prioritize regions with rich information, we propose deforming regions that are *less* semantically rich. Our deformation strategy helps the model learn to extract features from the most informative or salient regions of a point cloud.

To evaluate the richness of semantics, we propose using curvature diversity as a measurement. For the curvature diversity-driven deformation, we adopt the following steps. First, we use Farthest Point Sampling (FPS) Moenning & Dodgson (2003) to sample $k$ center points as centers of $k$ regions of a given point cloud. Then, for each center point, we use $k$-Nearest Neighbor ($k$-NN) to select $m$ nearest points. As a result, each region is represented by a center point along with these $m$ nearest points. Next, we select the $N$ regions with

the *smallest* curvature diversity to deform. To deform these selected regions, we replace all the points within these regions with new points. These new points are generated by sampling from a Gaussian distribution, where the mean is set to the average position of all the original points in that region, and the variance is set to 0.001. In Figure 1, $\hat{X}_s$ and $\hat{X}_t$ represent the deformed samples, and the points shown in grayscale are those drawn from the Gaussian distribution.

We compute point curvature using PCA Zou et al. (2021). Specifically, we first select a small neighborhood around each point and apply PCA to determine the principal directions and their eigenvalues. The curvature is then calculated as $c = |\lambda_{\min}| / \sum_{i=1}^{K} |\lambda_i|$, where $\lambda_{\min}$ is the smallest eigenvalue of the matrix, and $K$ is the number of eigenvalues. Larger variation in curvature indicates a more intricate geometry and more significant shape changes within a region. The fourth lamp sample in the bottom row of Figure 1 illustrates this property: regions with warmer colors represent areas of higher curvature diversity. To measure the diversity or variation of curvature in a region, we propose to use *entropy of curvature*. Entropy effectively captures the variations in curvature values, allowing quantifying the richness of semantics within a region. Formally, we use the following to measure the curvature diversity:

$$
c_{\min}^j = \min_{c_i^j \in R^j} \{c_i^j\}_{i=1}^{N_{R^j}}, \quad c_{\max}^j = \max_{c_i^j \in R^j} \{c_i^j\}_{i=1}^{N_{R^j}},
$$
$$
c_{i,\text{norm}}^j = \frac{c_i^j - c_{\min}^j}{c_{\max}^j - c_{\min}^j + 10^{-10}}, \quad H(c_{\text{norm}}^j) = -\sum_{i=1}^{N_{R^j}} c_{i,\text{norm}}^j \cdot \log(c_{i,\text{norm}}^j + 10^{-10}),
\tag{1}
$$

where, $c_i^j$ represents the curvatures of the $i$-th point in the $j$-th region of the point cloud which contains $N_{R^j}$ points in total. To standardize these values, we first calculate $c_{\min}^j$ and $c_{\max}^j$, which are the minimum and maximum values of all curvatures within the $j$-th region, respectively. Using these values, we then normalize the curvature values to be in $[0, 1]$, denoted as $\{c_{i,\text{norm}}^j\}$. Then, we calculate the curvature diversity $H(c_{\text{norm}}^j)$ by applying entropy.[1] The point-wise curvature values are computed only once before training. During training, we only compute the curvature diversity (entropy) for each region, based on these precomputed curvature values. This avoids repeated curvature estimation and keeps the runtime overhead minimal. Since curvature diversity calculation involves simple normalization and entropy over small regions, its cost is negligible compared to the main training process.

### 3.3 Deformation Reconstruction Loss

After deforming the selected regions, we obtain a deformed point cloud $x^d$ from the original $x$. The deformed input $x^d$ is processed by the feature extractor $E$ to generate $E(x^d)$, which is then passed to a reconstruction decoder $h_{\text{SSL}}$ to reconstruct $x$. The self-supervised loss $\mathcal{L}_{\text{SSL}}$ minimizes the distance between $h_{\text{SSL}}(E(x^d))$ and $x$. We use the Chamfer distance in $\mathcal{L}_{\text{SSL}}$, focusing on the original points in $x$ within the deformed region $R$ and their reconstructions from $x^d$. Formally, let $I \subset \{1, 2, \ldots\}$ represent the indices of the points in $x \cap R$, and we define $\mathcal{L}_{\text{SSL}}$ as follows:

$$
\mathcal{L}_{\text{SSL}} = \sum_{(x, x^d, I)} \left( \{x_i\}_{i \in I}, \{h_{\text{SSL}}(E(x^d))_i\}_{i \in I} \right),
\tag{2}
$$

where $x_i$ is the $i$-th point in the point cloud $x$ and $D$ denotes the Chamfer distance which is defined as:

$$
D(R_1, R_2) = \sum_{a \in R_1} \min_{b \in R_2} \|a - b\|_2^2 + \sum_{b \in R_2} \min_{a \in R_1} \|b - a\|_2^2,
\tag{3}
$$

where $D(R_1, R_2)$ measures the discrepancy between point cloud regions $R_1, R_2 \subset \mathbb{R}^3$. Note that when we reconstruct only the deformed regions, we can also reduce computational burden and improve time efficiency.

---

[1]Rigorously speaking, $\{c_{i,\text{norm}}^j\}$ is an un-normalized distribution without being divided by a partition function or normalization constant, but it does not affect our claim of curvature diversity.

### 3.4 Domain Alignment via D-NWD

We begin with introducing the 1-Wasserstein distance.

**Definition 1 (1-Wasserstein distance). Adler & Lunz (2018)** *The 1-Wasserstein distance quantifies the minimal cost of transporting mass between two probability measures defined on the same metric space. Let $\mu$ and $\nu$ be two probability measures over a metric space $(\Omega, d)$, where $d(x, y)$ denotes the distance between points $x$ and $y$ in $\Omega$. Then the 1-Wasserstein distance is defined as:*

$$W_1(\mu, \nu) = \inf_{\gamma \in \Gamma(\mu, \nu)} \int_{\Omega \times \Omega} d(x, y) \, d\gamma(x, y), \tag{4}$$

*where $\Gamma(\mu, \nu)$ is the set of all couplings with marginals $\mu$ and $\nu$; that is,*

$$\int_{\Omega} \gamma(x, y) \, dy = \mu(x), \quad \int_{\Omega} \gamma(x, y) \, dx = \nu(y).$$

*The Kantorovich–Rubinstein duality further states that the 1-Wasserstein distance can be expressed in a dual form as:*

$$W_1(\mu, \nu) = \sup_{\|h\|_L \leq K_L} \mathbb{E}_{x \sim \mu}[h(x)] - \mathbb{E}_{x \sim \nu}[h(x)], \tag{5}$$

*where the supremum is taken over all functions $h : \Omega \to \mathbb{R}$ with Lipschitz constant at most $K_L$, i.e.,*

$$\|h\|_L := \sup_{x \neq y} \frac{|h(x) - h(y)|}{d(x, y)} \leq K_L. \tag{6}$$

The Nuclear-norm Wasserstein Discrepancy (NWD) Chen et al. (2022) belongs to the family of 1-Wasserstein distances, with sophisticatedly chosen $h$'s. Below, we present the form of $h$ in NWD. Consider a prediction matrix $P \in \mathbb{R}^{b \times M}$ predicted by the classifier $C$, where $b$ is the number of samples in a batch and $M$ is the number of classes. The non-negative self-correlation matrix $Z \in \mathbb{R}^{M \times M}$ is computed as $Z = P^T P$. The intra-class correlation $I_a$ is defined as the sum of the main diagonal elements of $Z$, and the inter-class correlation $I_e$ is the sum of all the off-diagonal elements of $Z$:

$$I_a = \sum_{i=1}^{M} Z_{ii}, \quad I_e = \sum_{i \neq j}^{M} Z_{ij}.$$

In the source domain, $I_a$ is large, and $I_e$ is relatively small because most samples are correctly classified. Conversely, in the target domain, $I_a$ is small, and $I_e$ is relatively large due to the large error from the lack of supervised training on the target domain. Hence, $I_a - I_e$ can represent the discrepancy between the two domains, as $I_a - I_e$ is large for the source domain but small for the target domain. Note that $I_a = \|P\|_F^2$ can be represented as the squared Frobenius norm of $P$, and thus $I_a - I_e = 2\|P\|_F^2 - b$. [2] We can rewrite $P_s = C(F_s)$ and $P_t = C(F_t)$, where $F_s$ and $F_t$ are feature representation batches from the source and target domains, respectively. We find $\|C\|_F$ gives high scores to the source domain and low scores to the target domain. Thus, we can set $h$ in Eq. 5 to be $\|C\|_F$ and represent the domain discrepancy as:

$$W_F(\nu_s, \nu_t) = \sup_{\|\|C\|_F\|_L \leq K_L} \mathbb{E}_{F_s \sim \nu_s}[\|C(F_s)\|_F] - \mathbb{E}_{F_t \sim \nu_t}[\|C(F_t)\|_F],$$

where $\nu_s$ is the probability measure for features of samples in $\mathcal{S}$ and $\nu_t$ is the probability measure for features of samples in $\mathcal{T}$. To enhance prediction diversity, the Frobenius norm can be replaced with the nuclear norm, which maximizes the rank of $P$ while still being bounded by the Frobenius norm Chen et al. (2022). Thus, the domain discrepancy can be rewritten as:

$$W_N(\nu_s, \nu_t) = \sup_{\|\|C\|_*\|_L \leq K_L} \mathbb{E}_{F_s \sim \nu_s}[\|C(F_s)\|_*] - \mathbb{E}_{F_t \sim \nu_t}[\|C(F_t)\|_*]. \tag{7}$$

---

[2] We have $\sum_{j=1}^{M} Z_{i,j} = 1, \forall i \in \{1, \cdots, b\}$ and $j \in \{1, \cdots, M\}$, and thus $I_a + I_e = b$ Chen et al. (2022).

The Eq. 7 is the formal definition of NWD, where $C$ denotes the classifier, and $\|\cdot\|_*$ represents the nuclear norm. In the NWD paper, empirically, NWD can be approximate by $\mathcal{L}_{\text{NWD}}$:

$$\mathcal{L}_{\text{NWD}} = \frac{1}{B_s} \sum_{i=1}^{B_s} \|C(F_s^i)\|_* - \frac{1}{B_t} \sum_{i=1}^{B_t} \|C(F_t^i)\|_*$$

where $F_s^i \sim \nu_s$ are the features for the $i$-th source batch and $F_t^i \sim \nu_t$ are the features for the $i$-th target batch. $B_s$ is the number of training batches in the source domain and $B_t$ is the number of training batches in the target domain.

$$\min_E \max_C \mathcal{L}_{\text{NWD}} \tag{8}$$

Then, the domain alignment is achieved through a min-max game presented in Eq. 8.

Now, we introduce our second component, Deformation-based Nuclear-norm Wasserstein Discrepancy (D-NWD). From the previous sections, the curvature diversity-driven deformation reconstruction helps reduce the domain gap between the source and target domains. To further complete classification or segmentation tasks in the presence of the domain gap, we propose D-NWD to align domains, as inspired by NWD Chen et al. (2022). Our D-NWD objective is defined as:

$$W_N(\nu_{s \cup s^d}, \nu_{t \cup t^d}) = \sup_{\||C\|_*\|_L \leq K_L} \mathbb{E}_{\hat{F}_s \sim \nu_{s \cup s^d}} [\|C(\hat{F}_s)\|_*] - \mathbb{E}_{\hat{F}_t \sim \nu_{t \cup t^d}} [\|C(\hat{F}_t)\|_*], \tag{9}$$

Here, $\nu_{s \cup s^d}$ and $\nu_{t \cup t^d}$ are probability measures defined over $\Omega_o \cup \Omega_d$, for the features from samples in the original and deformed source and target domains. We align the probability measure of features from the original and deformed samples in the source domain with that of the target domain. Our motivation is that taking features from deformed samples into account would provide a richer, more robust feature space, reduce overfitting, and increase the model's adaptability to variations inherent in data. This technique differs from using NWD, which aligns $\nu_s$ and $\nu_t$ defined over $\Omega_o$, the probability measures for the features from samples in the original source and target domains. Empirically, our objective in Eq. 9 be approximated by $\mathcal{L}_{\text{D-NWD}}$:

$$\mathcal{L}_{\text{D-NWD}} = \frac{1}{B_s} \sum_{i=1}^{B_s} \|C(\hat{F}_s^i)\|_* - \frac{1}{B_t} \sum_{i=1}^{B_t} \|C(\hat{F}_t^i)\|_*, \tag{10}$$

$\hat{F}_s^i \sim \nu_{s \cup s^d}$ represents the features for the $i$-th source batch and $\hat{F}_t^i \sim \nu_{t \cup t^d}$ represents the features for the $i$-th target batch. In practice, we obtain the original and deformed samples by first sampling from the original domain and then generating the corresponding deformed versions. The alignment is then performed through a min-max game, described in the following:

$$\min_E \max_C \mathcal{L}_{\text{D-NWD}}. \tag{11}$$

To avoid alternating updates, we employ a Gradient Reverse Layer Ganin et al. (2016), following the approach in Chen et al. (2022), to make the learned features discriminative and domain-agnostic.

### 3.5 Overall Loss

In addition to deformation and domain alignment loss defined in Eq. 2 and Eq. 11, we use a cross-entropy loss $\mathcal{L}_{\text{CLS}}$ on both the original and the deformed source domain samples for supervised training of the classifier:

$$\mathcal{L}_{\text{CLS}} = \frac{1}{B_s} \sum_{i=1}^{B_s} \mathcal{L}_{\text{CE}}(C(\hat{F}_s^i), Y_s^i), \tag{12}$$

where $Y_s^i$ are labels for batch $\hat{F}_s^i$. Since we have no access to the ground-truth labels for the target domain data, it is impossible to use the supervised cross-entropy loss as in Eq. 12 on samples from $\mathcal{T}$ and $\widetilde{\mathcal{T}}$. One straightforward alternative is to adopt pseudo-labels as in Fan et al. (2022); Liang et al. (2022); Zou et al. (2021); Shen et al. (2022). However, this strategy has the risk that the classifier might mistakenly predict

target samples as the major classes of the source domain. Instead, we use NWD to ensure consistency in predictions between $\mathcal{T}$ and $\mathcal{T}^d$. Thus, we define a target domain loss $\mathcal{L}_{\mathrm{NWD}}^{\mathcal{T}}$ as:

$$\mathcal{L}_{\mathrm{NWD}}^{\mathcal{T}} = \frac{1}{B_t} \sum_{i=1}^{B_t} \|C(F_t^i)\|_* - \frac{1}{B_t} \sum_{i=1}^{B_t} \|C(F_{t^d}^i)\|_*, \tag{13}$$

where $F_{t^d}^i \sim \nu_{t^d}$ denotes the deformed target domain batch and $F_t^i \sim \nu_t$ denotes the original target domain batch. Combining Eq. 2, Eq. 12, Eq. 11 and Eq. 13 together, our overall objective loss is:

$$\begin{aligned} \min_{E, h_{\mathrm{SSL}}, C} & \alpha \mathcal{L}_{\mathrm{CLS}} + \gamma \mathcal{L}_{\mathrm{SSL}}, \\ \min_E \max_C & \beta_1 \mathcal{L}_{\mathrm{D\text{-}NWD}} + \beta_2 \mathcal{L}_{\mathrm{NWD}}^{\mathcal{T}}, \end{aligned} \tag{14}$$

where $\alpha, \gamma, \beta_1, \beta_2$ can be tuned using the target domain validation set and setting details can be found in Appendix A.2. Note that $\mathcal{L}_{\mathrm{D\text{-}NWD}}$ and $\mathcal{L}_{\mathrm{NWD}}^{\mathcal{T}}$ serve distinct and complementary purposes in Equation 14. $\mathcal{L}_{\mathrm{D\text{-}NWD}}$ is specifically used to minimize the domain gap between source and target features, while $\mathcal{L}_{\mathrm{NWD}}^{\mathcal{T}}$ is used to improve prediction consistency between original and deformed target distributions, functioning similarly to a cross-entropy loss between pseudo-labels and predicted labels on target samples. We use NWD over pseudo-labeling since the latter can reinforce incorrect predictions when the classifier is biased toward majority source domain classes.

## 4 Theoretical Analysis

In this section, we present our theoretical contribution for D-NWD, demonstrating the effectiveness of extending NWD to both original and deformed samples. We provide new bounds in Theorems 1 and 2 along with their non-trivial proofs. Following Ben-David et al. (2006) and Chen et al. (2022), we perform our analysis in a binary classification scenario, which can be easily adapted to multi-class classification through reduction techniques such as one-vs-all Rifkin & Klautau (2004) or one-vs-one Allwein et al. (2000) approaches. Consider $\{C : \mathbb{R}^n \to [0,1]\}$ as a set of source classifiers within the hypothesis space $\mathcal{H}$. Let $\nu_s$ defined on $\Omega_o$ be the probability measure of original source domain and $\nu_{s_d}$ defined on $\Omega_d$ be the probability measure of deformed source domain. We define $\nu_t$ and $\nu_t^d$ in a similar way. The risk or error of classifier $C$ on the original source domain is defined as $\varepsilon_s(C) = \mathbb{E}_{f_s \sim \nu_s}[|C(f_s) - y_s|]$, where $y_s$ is the label associated with the feature $f_s$. We then define $\varepsilon_{s \cup s^d}(C) = \mathbb{E}_{\hat{f}_s \sim \nu_{s \cup s^d}}[|C(\hat{f}_s) - \hat{y}_s|]$, where $\hat{y}_s$ is the label associated with $\hat{f}_s$. Similarly, we define $\varepsilon_t(C), \varepsilon_{t \cup t^d}(C)$ as the errors on the target domain. The optimal classifier is defined as $C^* = \arg\min_C \varepsilon_{s \cup s^d}(C) + \varepsilon_t(C)$ which minimizes the combined error across $\nu_{s \cup s^d}$ and $\nu_t$. Our Theorem 1 demonstrates that the expected target error $\varepsilon_t(C)$ can be bounded by the D-NWD on $\nu_{s \cup s^d}$ and $\nu_{t \cup t^d}$, $W_N(\nu_{s \cup s^d}, \nu_{t \cup t^d})$. Building on Theorem 1, we derive Theorem 2. Theorem 2 establishes that $\varepsilon_t(C)$ can be bounded by D-NWD on empirical probability measures $\hat{\nu}_{s \cup s^d}$ and $\hat{\nu}_{t \cup t^d}$, $W_N(\hat{\nu}_{s \cup s^d}, \hat{\nu}_{t \cup t^d})$. **All proofs can be found in Appendix A.1.**

**Theorem 1.** *Let* $(\Omega_o, \mathcal{F}_o, \nu_s)$, $(\Omega_d, \mathcal{F}_d, \nu_{s^d})$, $(\Omega_o, \mathcal{F}_o, \nu_t)$, *and* $(\Omega_d, \mathcal{F}_d, \nu_{t^d})$ *be four probability spaces, where* $\Omega_o$ *and* $\Omega_d$ *are disjoint and* $\Omega_o \cup \Omega_d \subseteq \mathbb{R}^n$. *With the results of Lemma 1, let* $(\Omega_o \cup \Omega_d, \mathcal{F}_u, \nu_{s \cup s^d})$ *and* $(\Omega_o \cup \Omega_d, \mathcal{F}_u, \nu_{t \cup t^d})$ *be two probability spaces with probability measures defined as* $\nu_{s \cup s^d} = 1/2\nu_s + 1/2\nu_{s^d}$ *and* $\nu_{t \cup t^d} = 1/2\nu_t + 1/2\nu_{t^d}$. *Specifically, when sampling from* $\nu_{t \cup t^d}$, *there is an equal probability of* $1/2$ *to sample from* $\nu_t$ *or* $\nu_{t^d}$. *Similarly, sampling from* $\nu_{s \cup s^d}$ *gives an equal probability of* $1/2$ *to draw from* $\nu_s$ *or* $\nu_{s^d}$. *Consider a classifier* $C \in \mathcal{H}_1$ *and an ideal classifier* $C^* = \arg\min_C \varepsilon_{s \cup s^d}(C) + \varepsilon_t(C)$ *satisfying the* $K_L$-*Lipschitz constraint, where* $\mathcal{H}_1$ *is a subspace of the hypothesis space* $\mathcal{H}$. *For every classifier* $C$ *in* $\mathcal{H}_1$, *the following inequality holds:*

$$\varepsilon_t(C) \leq 2\varepsilon_{s \cup s^d}(C) + 4K_L \cdot W_N(\nu_{s \cup s^d}, \nu_{t \cup t^d}) + \eta^*, \tag{15}$$

*where* $\eta^* = 2\varepsilon_{s \cup s^d}(C^*) + \varepsilon_t(C^*)$ *is the ideal combined error and is a sufficiently small constant.*

**Theorem 2.** *Under the assumption of Theorem 1,* $\Omega_o$ *and* $\Omega_d$ *are disjoint and* $\Omega_o \cup \Omega_d \subseteq \mathbb{R}^n$. *Let*

$(\Omega_o \cup \Omega_d, \mathcal{F}_u, \nu_{s \cup s^d})$ and $(\Omega_o \cup \Omega_d, \mathcal{F}_u, \nu_{t \cup t^d})$ be two probability spaces with $\nu_{s \cup s^d} = 1/2\nu_s + 1/2\nu_{s^d}$ and $\nu_{t \cup t^d} = 1/2\nu_t + 1/2\nu_{t^d}$, where $\nu_s, \nu_{s^d}, \nu_t, \nu_{t^d}$ each has a square-exponential moment. From Lemma 3 and 4, $\nu_{s \cup s^d}$ satsifies $T_1(\eta_s)$ for some $\eta_s$ and $\nu_{t \cup t^d}$ satsifies $T_1(\eta_t)$ for some $\eta_t$. Let $\{\hat{f}_s^i\}_{i=1}^{N_s}$ and $\{\hat{f}_t^i\}_{i=1}^{N_t}$ be two sample sets of size $N_s$ and $N_t$ drawn i.i.d from $\nu_{s \cup s^d}$ and $\nu_{t \cup t^d}$, respectively. $\hat{\nu}_{s \cup s^d} = \frac{1}{N_s} \sum_{i=1}^{N_s} \delta_{\hat{f}_s^i}$ and $\hat{\nu}_{t \cup t^d} = \frac{1}{N_t} \sum_{i=1}^{N_t} \delta_{\hat{f}_t^i}$ are associated empirical probability measures. Then, for any $n' > n$ and $\eta' < \min(\eta_s, \eta_t)$, there exists a constant $N_0$ depending on $n'$ such that for any $\delta > 0$ and $\min(N_s, N_t) \geq N_0 \max(\delta^{-(n'+2)}, 1)$, with probability at least $1 - \delta$, the following holds for all $C$:

$$\varepsilon_t(C) \leq 2\varepsilon_{s \cup s^d}(C) + 4K_L \cdot W_N(\hat{\nu}_{s \cup s^d}, \hat{\nu}_{t \cup t^d}) + \eta^* + 4K_L \cdot \sqrt{\frac{2}{\eta'} \log \frac{1}{\delta}} \left( \sqrt{\frac{1}{N_s}} + \sqrt{\frac{1}{N_t}} \right), \qquad (16)$$

where $\eta^* = 2\varepsilon_{s \cup s^d}(C^*) + \varepsilon_t(C^*)$ is the ideal combined error and is a sufficiently small constant.

Equation 16 justifies why our method can be effective empirically. Specifically, $\eta^*$ are sufficiently small constants for relevant domains with consistent labels because it is the error corresponding to the ideal classifier $C^*$. The term $\sqrt{\frac{2}{\eta'} \log \frac{1}{\delta}} \left( \sqrt{\frac{1}{N_s}} + \sqrt{\frac{1}{N_t}} \right)$ is also a small constant when training dataset sizes, $N_s$ and $N_t$, are large. $\varepsilon_{s \cup s^d}(C)$ is a supervised classification loss, since source domain samples have labels. Therefore, the primary objective is to minimize $W_N(\hat{\nu}_{s \cup s^d}, \hat{\nu}_{t \cup t^d})$, our D-NWD on empirical measures $\hat{\nu}_{s \cup s^d}$ and $\hat{\nu}_{t \cup t^d}$. Hence, minimizing $W_N(\hat{\nu}_{s \cup s^d}, \hat{\nu}_{t \cup t^d})$ can reduce error on the original target domain $\mathcal{T}$ and improve the model's performance on the target domain. It is important to note that our theoretical analysis is not intended to prove that D-NWD is superior to NWD. A direct comparison between NWD and D-NWD is not feasible, as they apply to different probability measures: $\hat{\nu}_{s \cup s^d}, \hat{\nu}_{t \cup t^d}$ for D-NWD and $\hat{\nu}_s, \hat{\nu}_t$ for NWD. Our theoretical contribution lies in showing that, regardless of the deformation method used, optimizing D-NWD on $\hat{\nu}_{s \cup s^d}$ and $\hat{\nu}_{t \cup t^d}$ can effectively reduce error on samples from $\mathcal{T}$. In other words, D-NWD mitigates the negative effects of domain gaps and enhances performance on the original target domain $\mathcal{T}$. Note that we use equal mixture weights—specifically, a 1:1 ratio—in the probability measure (e.g., $\frac{1}{2}\nu_s + \frac{1}{2}\nu_s^d$) to simplify the implementation. However, this assumption is not essential for the validity of our conclusions. The theoretical results can be generalized to arbitrary sampling ratios by adjusting the mixture weights in the probability measures accordingly. In such cases, the only change would be in the constant factors reflecting the updated contribution from each component in the mixture. The core theoretical framework and its implications remain intact. In Appendix A.3, we present the experiment results with different ratios.

## 5 Experiments

We evaluate our method on the **PointDA-10** Qin et al. (2019) dataset, a domain adaptation dataset for point cloud classification, and on **PointSegDA** Achituve et al. (2021), a dataset for point cloud segmentation. For the PointDA-10 dataset, we compare our approach against the state-of-the-art methods for point cloud domain adaptation, including **DANN** Ganin et al. (2016), **PointDAN** Qin et al. (2019), **RS** Sauder & Sievers (2019), **DefRec+PCM** Achituve et al. (2021), **GAST** Zou et al. (2021), **ImplicitPCDA** Shen et al. (2022), and the most recent method with publicly available codes, **PCFEA** Wang & el al (2025). Note that many recent methods do not provide codes for reproducibility, which limits their inclusion. For the **PointSegDA** dataset, we compare our method with **RS**, **DefRec+PCM**, **GAST**, **ImplicitPCDA**, and **Adapt-SegMap** Tsai et al. (2018). Unfortunately, no suitable methods from the past 1–2 years with available codes are applicable to this dataset. For both datasets, we also evaluate two upper bounds: **Supervised-T**, which involves training exclusively on labeled target samples, and **Supervised**, which uses both labeled source and target samples. Additionally, we assess a lower bound, **Unsupervised**, which utilizes only labeled source samples.

Additionally, for the PointDA-10 dataset, we incorporate Self-Paced Self-Training (SPST) into GAST, ImplicitPCDA, and our method, as SPST is originally included in both GAST and ImplicitPCDA. We exclude SPST for the PointSegDA dataset. SPST typically relies on ranking training samples by difficulty and gradually incorporating harder examples into training. The training samples for point cloud segmentation tasks are points in point clouds. However, mIoU is a global metric that evaluates performance across an entire point cloud, making it challenging to assign difficulty scores to individual points in a point cloud. The

mechanism of SPST mismatches the per-point cloud, rather than per-point, evaluation criterion of mIoU. **Hyperparameter settings and implementation details can be found in Appendix A.2.**

### 5.1 Datasets

We use **PointDA-10** and **PointSegDA** datasets in our experiments. **PointDA-10** consists of three domains: ShapeNet-10 Chang et al. (2015), ModelNet-10 Wu et al. (2015), and ScanNet-10 Dai et al. (2017), each sharing ten distinct classes. **PointSegDA** consists of four domains: ADOBE, FAUST, MIT, and SCAPE. These domains share eight distinct classes of human body parts but vary in point distribution, pose, and scanned humans.

### 5.2 Training Scheme

We use DGCNN as the feature extractor Achituve et al. (2021) for fair comparison. We repeat our experiments three times using distinct random seeds for initialization and report the average accuracy and standard deviation. To ensure a fair comparison, we maintain the same seed for data shuffling and use the Adam optimizer Kingma & Ba (2014) for optimization.

| Models | MS | MS$^+$ | SM | SS$^+$ | S$^+$M | S$^+$S | Avg |
|---|---|---|---|---|---|---|---|
| Supervised-T | $93.9_{\pm0.2}$ | $78.4_{\pm0.6}$ | $96.2_{\pm0.1}$ | $78.4_{\pm0.6}$ | $96.2_{\pm0.4}$ | $93.9_{\pm0.2}$ | 89.5 |
| Unsupervised | $83.3_{\pm0.7}$ | $43.8_{\pm2.3}$ | $75.5_{\pm1.8}$ | $42.5_{\pm1.4}$ | $63.8_{\pm3.9}$ | $64.2_{\pm0.8}$ | 62.2 |
| DANN Ganin et al. (2016) | $75.3_{\pm0.6}$ | $41.5_{\pm0.2}$ | $62.5_{\pm1.4}$ | $46.1_{\pm2.8}$ | $53.3_{\pm1.2}$ | $63.2_{\pm1.2}$ | 57.0 |
| PointDAN Qin et al. (2019) | $82.5_{\pm0.8}$ | $47.7_{\pm1.0}$ | $77.0_{\pm0.3}$ | $48.5_{\pm2.1}$ | $55.6_{\pm0.6}$ | $67.2_{\pm2.7}$ | 63.1 |
| RS Sauder & Sievers (2019) | $81.5_{\pm1.2}$ | $35.2_{\pm5.9}$ | $71.9_{\pm1.4}$ | $39.8_{\pm0.7}$ | $61.0_{\pm3.3}$ | $63.6_{\pm3.4}$ | 58.8 |
| DefRec+PCM Achituve et al. (2021) | $81.7_{\pm0.6}$ | $51.8_{\pm0.3}$ | $78.6_{\pm0.7}$ | $54.5_{\pm0.3}$ | $73.7_{\pm1.6}$ | $71.1_{\pm1.4}$ | 68.6 |
| PCFEA Wang & el al (2025) | $84.4_{\pm1.1}$ | $47.9_{\pm2.9}$ | $71.8_{\pm1.9}$ | $47.1_{\pm0.7}$ | $71.8_{\pm7.4}$ | $68.1_{\pm2.1}$ | 65.2 |
| GAST Zou et al. (2021) | $82.3_{\pm0.6}$ | $53.0_{\pm1.1}$ | $72.6_{\pm1.9}$ | $47.6_{\pm1.5}$ | $64.6_{\pm1.5}$ | $66.8_{\pm0.6}$ | 64.5 |
| GAST+SPST | $84.5_{\pm0.5}$ | $54.1_{\pm1.8}$ | $80.1_{\pm4.6}$ | $46.7_{\pm0.6}$ | $81.5_{\pm1.7}$ | $66.7_{\pm1.1}$ | 68.9 |
| ImplicitPCDA Shen et al. (2022) | $79.5_{\pm0.4}$ | $41.7_{\pm1.3}$ | $72.9_{\pm1.0}$ | $47.5_{\pm2.9}$ | $67.6_{\pm5.2}$ | $66.4_{\pm0.9}$ | 62.6 |
| ImplicitPCDA+SPST | $81.3_{\pm2.2}$ | $33.2_{\pm13.4}$ | $73.2_{\pm3.4}$ | $38.0_{\pm4.6}$ | $66.9_{\pm7.7}$ | $75.0_{\pm2.7}$ | 61.3 |
| CDND | $84.1_{\pm0.3}$ | $\mathbf{58.7}_{\pm0.8}$ | $76.2_{\pm0.0}$ | $\mathbf{55.7}_{\pm1.0}$ | $75.1_{\pm1.5}$ | $72.0_{\pm1.9}$ | 70.3 |
| CDND+SPST | $\mathbf{85.4}_{\pm1.1}$ | $57.6_{\pm1.3}$ | $\mathbf{85.0}_{\pm2.2}$ | $54.5_{\pm1.1}$ | $\mathbf{82.6}_{\pm0.7}$ | $\mathbf{74.6}_{\pm4.4}$ | $\mathbf{73.3}$ |

Table 1: Performance results (accuracy) on PointDA-10 dataset.

| Models | MS | MS$^+$ | SM | SS$^+$ | S$^+$M | S$^+$S | Avg |
|---|---|---|---|---|---|---|---|
| NWD | $83.3_{\pm0.7}$ | $46.7_{\pm1.7}$ | $75.5_{\pm1.8}$ | $48.9_{\pm2.5}$ | $63.8_{\pm3.9}$ | $66.7_{\pm1.9}$ | 64.2 |
| DefRec | $83.4_{\pm0.5}$ | $46.9_{\pm2.3}$ | $74.5_{\pm0.9}$ | $46.3_{\pm0.6}$ | $67.7_{\pm2.3}$ | $64.0_{\pm0.8}$ | 64.0 |
| DefRec+NWD | $83.4_{\pm0.5}$ | $51.2_{\pm3.0}$ | $74.5_{\pm0.9}$ | $53.7_{\pm3.8}$ | $67.7_{\pm2.3}$ | $68.5_{\pm2.4}$ | 66.5 |
| DefRec+D-NWD | $83.4_{\pm0.5}$ | $53.1_{\pm2.3}$ | $74.5_{\pm0.9}$ | $54.6_{\pm1.0}$ | $67.7_{\pm2.3}$ | $67.4_{\pm0.1}$ | 66.8 |
| CurvRec(S)-High | $83.8_{\pm0.9}$ | $52.0_{\pm1.4}$ | $\mathbf{78.0}_{\pm1.0}$ | $45.9_{\pm3.8}$ | $72.5_{\pm1.4}$ | $66.7_{\pm1.1}$ | 66.5 |
| CurvRec(S)-Low | $83.1_{\pm0.9}$ | $53.0_{\pm1.9}$ | $74.9_{\pm0.8}$ | $44.7_{\pm1.2}$ | $74.8_{\pm0.9}$ | $65.9_{\pm0.2}$ | 66.1 |
| CurvRec(En)-High | $82.9_{\pm1.5}$ | $52.1_{\pm0.4}$ | $77.0_{\pm0.3}$ | $46.7_{\pm1.0}$ | $70.9_{\pm0.6}$ | $65.8_{\pm0.4}$ | 65.9 |
| CurvRec | $\mathbf{84.1}_{\pm0.3}$ | $52.2_{\pm1.3}$ | $76.2_{\pm0.0}$ | $50.1_{\pm0.3}$ | $\mathbf{75.1}_{\pm1.5}$ | $66.4_{\pm1.5}$ | 67.4 |
| CurvRec+PCM | $83.0_{\pm0.5}$ | $53.7_{\pm1.0}$ | $74.0_{\pm0.6}$ | $54.8_{\pm1.1}$ | $73.8_{\pm1.1}$ | $\mathbf{76.8}_{\pm0.9}$ | 69.4 |
| CurvRec+NWD | $\mathbf{84.1}_{\pm0.2}$ | $54.3_{\pm2.2}$ | $76.2_{\pm0.0}$ | $52.7_{\pm2.1}$ | $\mathbf{75.1}_{\pm1.5}$ | $70.6_{\pm2.2}$ | 68.8 |
| CDND, $\beta_2 = 0$ | $84.1_{\pm0.3}$ | $57.4_{\pm1.2}$ | $76.2_{\pm0.0}$ | $\mathbf{55.7}_{\pm0.2}$ | $75.1_{\pm1.5}$ | $69.9_{\pm1.7}$ | 69.7 |
| CDND (CurvRec+D-NWD) | $\mathbf{84.1}_{\pm0.3}$ | $\mathbf{58.7}_{\pm0.8}$ | $76.2_{\pm0.0}$ | $\mathbf{55.7}_{\pm1.0}$ | $\mathbf{75.1}_{\pm1.5}$ | $72.0_{\pm1.9}$ | $\mathbf{70.3}$ |

Table 2: Ablation study results (accuracy) on PointDA-10 dataset.

| Models | FA | FM | FS | MA | MF | MS | AF | AM | AS | SA | SF | SM | AVG |
|---|---|---|---|---|---|---|---|---|---|---|---|---|---|
| Supervised | $80.9_{\pm7.2}$ | $81.8_{\pm0.3}$ | $82.4_{\pm1.2}$ | $80.9_{\pm7.2}$ | $84.0_{\pm1.8}$ | $82.4_{\pm1.2}$ | $84.0_{\pm1.8}$ | $81.8_{\pm0.3}$ | $82.4_{\pm1.2}$ | $80.9_{\pm7.2}$ | $84.0_{\pm1.8}$ | $81.8_{\pm0.3}$ | 82.3 |
| Unsupervised | $78.5_{\pm0.4}$ | $60.9_{\pm0.6}$ | $66.5_{\pm0.6}$ | $26.6_{\pm3.5}$ | $33.6_{\pm1.3}$ | $69.9_{\pm1.2}$ | $38.5_{\pm2.2}$ | $31.2_{\pm1.4}$ | $30.0_{\pm3.6}$ | $74.1_{\pm1.0}$ | $68.4_{\pm2.4}$ | $64.5_{\pm0.5}$ | 53.6 |
| AdaptSegMap | $70.5_{\pm3.4}$ | $60.1_{\pm0.6}$ | $65.3_{\pm1.3}$ | $49.1_{\pm9.7}$ | $54.0_{\pm0.5}$ | $62.8_{\pm7.6}$ | $\mathbf{44.2}_{\pm1.7}$ | $35.4_{\pm0.3}$ | $35.1_{\pm1.4}$ | $70.1_{\pm2.5}$ | $67.7_{\pm1.4}$ | $63.8_{\pm1.2}$ | 56.5 |
| RS | $78.7_{\pm0.5}$ | $60.7_{\pm0.4}$ | $\mathbf{66.9}_{\pm0.4}$ | $59.6_{\pm5.0}$ | $38.4_{\pm2.1}$ | $\mathbf{70.4}_{\pm1.0}$ | $44.0_{\pm0.6}$ | $30.4_{\pm0.5}$ | $36.6_{\pm0.8}$ | $70.7_{\pm0.8}$ | $\mathbf{73.0}_{\pm1.5}$ | $\mathbf{65.3}_{\pm1.3}$ | 57.9 |
| DefRec+PCM | $78.8_{\pm0.2}$ | $\mathbf{60.9}_{\pm0.8}$ | $63.6_{\pm0.1}$ | $48.1_{\pm0.4}$ | $48.6_{\pm2.4}$ | $70.1_{\pm0.8}$ | $46.9_{\pm1.0}$ | $33.2_{\pm0.3}$ | $37.6_{\pm0.1}$ | $66.3_{\pm1.7}$ | $66.5_{\pm1.0}$ | $62.6_{\pm0.2}$ | 56.9 |
| GAST | $76.7_{\pm2.3}$ | $55.0_{\pm1.0}$ | $60.3_{\pm1.0}$ | $52.1_{\pm4.4}$ | $35.2_{\pm0.4}$ | $69.6_{\pm1.2}$ | $43.3_{\pm3.7}$ | $25.9_{\pm3.6}$ | $30.8_{\pm4.0}$ | $57.4_{\pm10.6}$ | $66.1_{\pm1.3}$ | $64.6_{\pm0.5}$ | 53.1 |
| ImplicitPCDA | $47.5_{\pm0.6}$ | $53.2_{\pm1.0}$ | $54.2_{\pm3.4}$ | $51.1_{\pm1.6}$ | $\mathbf{64.0}_{\pm1.3}$ | $56.1_{\pm4.2}$ | $44.1_{\pm0.9}$ | $\mathbf{42.3}_{\pm1.3}$ | $\mathbf{40.5}_{\pm1.2}$ | $49.7_{\pm2.1}$ | $70.6_{\pm1.4}$ | $55.0_{\pm2.5}$ | 52.4 |
| CDND | $\mathbf{81.5}_{\pm2.0}$ | $60.7_{\pm0.5}$ | $61.4_{\pm0.5}$ | $\mathbf{68.6}_{\pm1.4}$ | $47.2_{\pm1.4}$ | $67.7_{\pm1.4}$ | $43.6_{\pm0.5}$ | $35.3_{\pm2.2}$ | $40.1_{\pm1.5}$ | $\mathbf{77.5}_{\pm0.5}$ | $70.4_{\pm1.1}$ | $65.1_{\pm0.3}$ | $\mathbf{59.9}$ |

Table 3: Performance results (mIoU) on PointSegDA dataset.

## 5.3 Comparative Results

**Results on PointDA.** The results are presented in Table 1. We use $S^+$ to represent the ScanNet dataset, M to represent ModelNet, and S to represent the ShapeNet dataset. MS, SM, etc. are the abbreviated source and target domain pairs. The CDND model shows significant improvement over the other approaches on the PointDA-10 dataset with the highest average accuracy of 70.3%, outperforming all other models. CDND delivers state-of-the-art performance on five out of six tasks. It excels in tasks with a large domain gap, such as MS$^+$, S$^+$M, SS$^+$, and S$^+$S. In these tasks, one domain is a synthetic dataset, and the other domain is a real-world dataset. This shows its proficiency in handling complex transformations. Especially, CDND scores 58.7% on MS$^+$, outperforming the second-best method by approximately 6%. Additionally, CDND maintains competitive accuracy in tasks with a small domain gap, such as MS and SM, with scores of 84.1% and 76.2%, respectively. With SPST, the performance is further improved, as CDND+SPST achieves 73.3%, outperforming GAST+SPST by 4.4% and ImplicitPCDA+SPST by 12%. Note that plain CDND also outperforms both GAST+SPST and ImplicitPCDA+SPST on average. The performance of CDND across various tasks highlights its ability to adapt to diverse domain challenges, making it a promising choice for point cloud classification in the UDA setting.

**Results on PointSegDA.** The results are presented in Table 3. We use A to represent the ADOBE dataset, F to represent the FAUST dataset, M to represent the MIT dataset, and S to represent the SCAPE dataset. AF, FA, etc. are the abbreviated source and target domain pairs. CDND achieves the highest average score of 59.9%, which surpasses the second-best method, RS, by a margin of 2.0%, which is significant in terms of mIoU on the segmentation task. Its superior performance is particularly evident in MA and SA tasks; in the MA task, CDND achieves a mIoU of 68.6%, outperforming RS by 9%. Similarly, in the SA task, CDND secures a mIoU of 77.5%, which is around 7% higher than RS. These results showcase its adaptability and learning capability. Additionally, in the FA task, CDND achieves a score of 81.5%, even slightly surpassing the supervised baseline. In other tasks, *i.e.*, FM, AS, and SM tasks, CDND either matches or comes very close to the top-performing models, validating its status as a consistently high-performing model. In Appendix A.3, we have visualization results to highlight the qualitative differences.

Overall, the performance improvement of our method, while consistent, is less pronounced compared to the results on PointDA-10. This is primarily due to the nature of the segmentation task, which involves fine-grained, per-point predictions. Our curvature diversity-driven deformation focuses on regional semantics and is more directly beneficial for object-level classification tasks. It is worth noting that other state-of-the-art methods also report smaller margins on PointSegDA. Nevertheless, our method still achieves the highest average mIoU, demonstrating its effectiveness in segmentation settings.

## 5.4 Ablation Study

To demonstrate the effectiveness of each component of CDND, we conduct ablative studies on the PointDA-10 dataset. There are several ways to evaluate curvature diversity. While standard deviation is commonly used to evaluate the diversity of data points, we propose using entropy. We compare our entropy-based approach (CurvRec(En)) with a standard deviation-based method (CurvRec(S)). To validate our hypothesis that focusing on low curvature diversity regions can improve performance, we investigate the impact of deform-

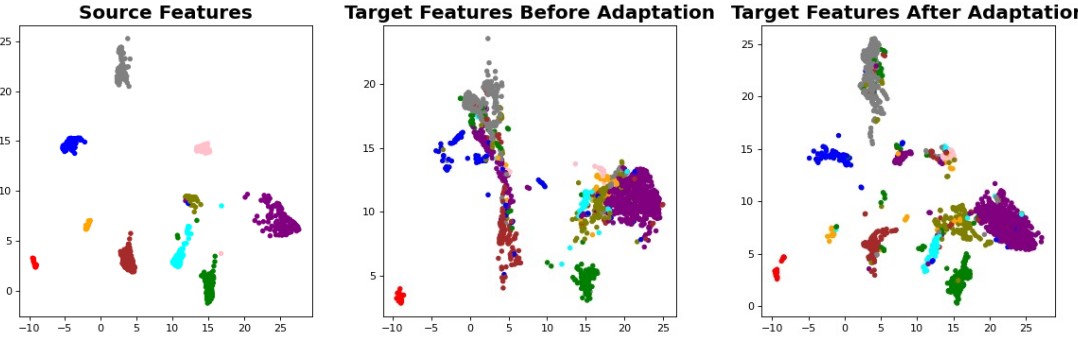

Figure 2: UMAP visualizations depict pre-activation data representations for the $MS^+$ task, with different colors denoting different classes. The center plot shows the target domain test data representations generated from a model trained on the source dataset without any adaptation. The left and right plots show the source and target domain data representations after adaptation using CDND.

ing areas with both high (CurvRec(En)+High, CurvRec(S)+High) and low (CurvRec, CurvRec(S)+Low) curvature diversity.

**Effectiveness of CurvRec.** We compare against CurvRec(En) variants with CurvRec(S) variants in Table 2. We observe a distinct difference between CurvRec(En)-High and CurvRec. In contrast, there is a much less distinction between CurvRec(S)-High and CurvRec(S)-Low. This observation suggests that entropy is a superior method for evaluating curvature diversity in regions. Notably, all CurvRec measures outperform DefRec, regardless of whether the focus is on high or low curvature diversity. However, CurvRec outperforms CurvRec(En)-High, showing that is is more effective to deform low curvature regions. Compared to the plain NWD, all CurvRec variants perform better than plain NWD. Specifically, CurvRec surpasses DefRec and NWD by approximately 3%. Though CurvRec demonstrates better performance overall, it does not outperform our proposed CDND. CDND (CurvRec +D-NWD), outperforms all CurvRec variants, DefRec variants, and plain NWD. This highlights the effectiveness of our D-NWD loss. Compared to CurvRec only, integrating with D-NWD improves average performance by 2.9%, with specific gains of 6.5% on $MS^+$ and 5.6% on $SS^+$.

**Effectiveness of D-NWD.** To illustrate the effectiveness of our D-NWD, we first compare CDND with two alternatives: CurvRec+PCM, which replaces D-NWD with PCM (PointMixup), and CurvRec+NWD. On average, CDND outperforms both methods. Specifically, compared to CurvRec+PCM, CDND achieves improvements of approximately 5% on $MS^+$, and 1% to 2% on $SS^+$, MS, and SM. When compared to CurvRec+NWD, CDND surpasses 4.4% on $MS^+$, 3% on $SS^+$, and 1.4% on $S^+S$. To further demonstrate the generalizability of D-NWD across different deformation methods, we include results for DefRec+D-NWD and DefRec+NWD. Compared to plain DefRec, DefRec+D-NWD achieves an overall improvement of 2.8%, with notable gains of 6.2% on $MS^+$, 3.4% on $S^+S$, and 8.3% on $SS^+$. Moreover, DefRec+D-NWD consistently outperforms DefRec+NWD on $MS^+$ and $S^+S$, as well as on average, although the margin is relatively small. Note that our theoretical analysis proves that D-NWD effectively reduces domain discrepancy across various deformation methods, but the degree to which it outperforms NWD depends on the deformation quality. With a well-designed deformations method like our proposed CurvRec, D-NWD can outperform NWD significantly. While D-NWD is crucial for tasks with large domain gaps (e.g., $MS^+$, $S^+S$, $SS^+$), its advantages are less significant in tasks with smaller domain gaps (e.g., MS, SM, $S^+M$), where CurvRec or DefRec alone already perform strongly. In these cases, plain CurvRec or DefRec performs better, so we retain their performance for MS, SM, and $S^+M$. Additionally, we test the use of $\mathcal{L}_{NWD}$ without adding $\mathcal{L}_{NWD}^{\mathcal{T}}$ to enhance consistency in the predictions of original target domain samples and deformed target domain samples, which corresponds to CDND with $\beta_2 = 0$. CDND with $\beta_2 = 0$ shows only a minor performance decrease compared to CDND, indicating that D-NWD is the primary contributor to good performance.

### 5.5 Analytic Experiments

We conduct analytical experiments to gain deeper insights into the effectiveness of our approach. Specifically, we assess how CDND impacts the *distribution* of the target domain in the classifier's output space for the challenging ModelNet to ScanNet task (MS$^+$); ModelNet is a synthetic dataset, while ScanNet is a real-world dataset, making the domain shift between them particularly challenging. We used UMAP to visualize and compare data representations of validation data from the source domain and test data from the target domain, both before and after applying CDND. Figure 2 shows each point as a data representation in the classifier's output space before softmax activation, with different colors denoting different classes. The middle plot in Figure 2 illustrates that, prior to adaptation, the classifier struggles with the target domain data, as points from different classes are heavily intermixed. However, after applying CDND, the class boundaries become more distinct, and the distribution of target domain representations aligns well with that of the source domain. This improvement is visible in the left and right plots of Figure 2, where the arrangement of points shows a more distinct pattern across both domains. In other words, we see that the feature space becomes domain-agnostic. This visualization demonstrates CDND's efficacy in reducing domain shift-induced performance degradation and enhancing class distinction. **More UMAP visualization analysis of other baselines can be found in Appendix A.3**.

## 6 Conclusion

We developed a new unsupervised domain adaptation approach for point cloud data. Our method integrates curvature diversity-based deformation with Deformation-based D-NWD to mitigate target domain performance degradation. Our theoretical analysis of D-NWD shows it minimizes an upper bound for target domain error. Additionally, we show that D-NWD can be applied to any deformation method. Experiments indicate that our approach surpasses SOTA methods on two point cloud benchmarks. Ablation studies confirm that both components of CDND are necessary for optimal performance. Future works include scenarios when source domain data is not accessible due to privacy or security concerns, or when the domains share only a subset of their classes.

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

# A    Appendix

## A.1    Proofs for Theorems

In this section, we first prove Theorem 1, which serves as the foundation for Theorem 2. Our proofs are structured as follows: we begin by proving Lemma 1, which supports a key assumption in Theorem 1. Next, we present the proof of Theorem 1. After proving Theorem 1, we prove Lemma 4 and conclude with the proof of Theorem 2.

**Definition 2 (Probability Spacce). Durrett (2019)** *A probability space is a triple $(\Omega, \mathcal{F}, \nu)$. $\Omega$ represents the sample space, the set of all possible outcomes. $\mathcal{F}$ represents the set of events and is a $\sigma$-algebra, which is a nonempty collection of subsets of $\Omega$. $\mathcal{F}$ is closed under complements and countable unions. $\nu$ represents a probability measure on the measurable space $(\Omega, \mathcal{F})$. It is a function $\nu : \mathcal{F} \to [0, 1]$ that assigns to each event $A \in \mathcal{F}$ a real value $\nu(A)$ (the probability of A). $\nu$ satisfies the following three axioms:*

- *Non-negativity: For every $A \in \mathcal{F}$, $\nu(A) \geq \nu(\emptyset) = 0$.*

- *Normalization: $\nu(\Omega) = 1$.*

- *$\sigma$-additivity (Countable Additivity): For any countable sequence of pairwise disjoint events $A_1, A_2, A_3, \dots \in \mathcal{F}$ (where $A_i \cap A_j = \emptyset$ for $i \neq j$),*

$$\nu \left( \bigcup_{i=1}^{\infty} A_i \right) = \sum_{i=1}^{\infty} \nu(A_i).$$

**Lemma 1.** *Let $(\Omega_1, \mathcal{F}_1, \nu_1)$ and $(\Omega_2, \mathcal{F}_2, \nu_2)$ be two probability spaces, where $\Omega_1, \Omega_2$ are two disjoint sample spaces. Let $p_1, p_2 \in [0, 1]$ be constants such that $p_1 + p_2 = 1$. Let $(\Omega_3, \mathcal{F}_3)$ be a measurable space, where $\mathcal{F}_3$ is the $\sigma$-algebra on $\Omega_3 = \Omega_1 \cup \Omega_2$. Then, the measure $\nu_3$ defined on the measurable space $(\Omega_3, \mathcal{F}_3)$ as:*

$$\nu_3(A) = p_1 \nu_1(A \cap \Omega_1) + p_2 \nu_2(A \cap \Omega_2), \quad \forall A \in \mathcal{F}_3,$$

*is a probability measure on* $(\Omega_3, \mathcal{F}_3)$.

*Proof.* Since $\nu_1$ and $\nu_2$ are probability measures, they satisfy $\nu_1(B) \geq 0$ for all $B \in \mathcal{F}_1$ and $\nu_2(C) \geq 0$ for all $C \in \mathcal{F}_2$. For any set $A \in \mathcal{F}_3$, we have:

$$\nu_3(A) = p_1\nu_1(A \cap \Omega_1) + p_2\nu_2(A \cap \Omega_2).$$

Given that $p_1, p_2 \geq 0$ and $\nu_1(A \cap \Omega_1) \geq 0$ and $\nu_2(A \cap \Omega_2) \geq 0$, it follows that $\nu(A) \geq 0$. Thus, $\nu$ is non-negative. Then, we need to show that $\nu(\Omega_1 \cup \Omega_2) = 1$. Consider:

$$\nu_3(\Omega_1 \cup \Omega_2) = p_1\nu_1((\Omega_1 \cup \Omega_2) \cap \Omega_1) + p_2\nu_2((\Omega_1 \cup \Omega_2) \cap \Omega_2).$$

Since $(\Omega_1 \cup \Omega_2) \cap \Omega_1 = \Omega_1$ and $(\Omega_1 \cup \Omega_2) \cap \Omega_2 = \Omega_2$, and $\nu_1(\Omega_1) = 1$ and $\nu_2(\Omega_2) = 1$, we have:

$$\nu_3(\Omega_1 \cup \Omega_2) = p_1 \cdot 1 + p_2 \cdot 1 = p_1 + p_2 = 1.$$

Thus, $\nu_3$ is normalized. Let $\{A_i\}_{i=1}^{\infty}$ be a countable collection of pairwise disjoint sets in $\mathcal{F}_3$. By definition of $\nu_3$,

$$\nu_3\left(\bigcup_{i=1}^{\infty} A_i\right) = p_1\nu_1\left(\left(\bigcup_{i=1}^{\infty} A_i\right) \cap \Omega_1\right) + p_2\nu_2\left(\left(\bigcup_{i=1}^{\infty} A_i\right) \cap \Omega_2\right).$$

Since the $A_i$ are pairwise disjoint, $(\bigcup_{i=1}^{\infty} A_i) \cap \Omega_1 = \bigcup_{i=1}^{\infty}(A_i \cap \Omega_1)$, and similarly for $\Omega_2$. Using the $\sigma$-additivity of $\nu_1$ and $\nu_2$:

$$p_1\nu_1\left(\bigcup_{i=1}^{\infty}(A_i \cap \Omega_1)\right) = p_1\sum_{i=1}^{\infty}\nu_1(A_i \cap \Omega_1),$$

$$p_2\nu_2\left(\bigcup_{i=1}^{\infty}(A_i \cap \Omega_2)\right) = p_2\sum_{i=1}^{\infty}\nu_2(A_i \cap \Omega_2).$$

Thus,

$$\nu_3\left(\bigcup_{i=1}^{\infty} A_i\right) = p_1\sum_{i=1}^{\infty}\nu_1(A_i \cap \Omega_1) + p_2\sum_{i=1}^{\infty}\nu_2(A_i \cap \Omega_2) = \sum_{i=1}^{\infty}\left(p_1\nu_1(A_i \cap \Omega_1) + p_2\nu_2(A_i \cap \Omega_2)\right).$$

Since $\nu_3(A_i) = p_1\nu_1(A_i \cap \Omega_1) + p_2\nu_2(A_i \cap \Omega_2)$, we get:

$$\nu_3\left(\bigcup_{i=1}^{\infty} A_i\right) = \sum_{i=1}^{\infty}\nu_3(A_i).$$

Thus, $\nu_3$ satisfies $\sigma$-additivity. Since $\nu_3$ satisfies non-negativity, normalization, and $\sigma$-additivity, by definition, $\nu_3$ is a valid probability measure.

Now, for the following Lemmas and Theorems, we define:

$$\varepsilon_s(C_1, C_2) = \mathbb{E}_{f_s \sim \nu_s}\left[|C_1(f_s) - C_2(f_s)|\right],$$

$$\varepsilon_{s \cup s^d}(C_1, C_2) = \mathbb{E}_{\hat{f}_s \sim \nu_{s \cup s^d}}\left[|C_1(\hat{f}_s) - C_2(\hat{f}_s)|\right].$$

where $C_1, C_2$ are two classifiers. We define $\varepsilon_t(C_1, C_2)$ and $\varepsilon_{t \cup t^d}(C_1, C_2)$ in the same manner.

**Lemma 2 (Lemma 1 Chen et al. (2022)).** *Let $\nu, \nu'$ be two probability measures on $(\Omega, \mathcal{F})$. Let $d(x, y)$ be the distance between $x \sim \nu$ and $y \sim \nu'$. $W_N$ represents the NWD. Given a family of classifiers $C \in \mathcal{H}_1$ and a ideal classifier $C^* \in \mathcal{H}_1$ satisfying the $K_L$-Lipschitz constraint, where $\mathcal{H}_1$ is a subspace of $\mathcal{H}$, the following holds for every $C, C^* \in \mathcal{H}_1$.*

$$|\varepsilon(C, C^*) - \varepsilon'(C, C^*)| \leq 2K_L \cdot W_N(\nu_1, \nu_2),$$

where $\varepsilon$ is the error on $\nu$ and $\varepsilon'$ is the error on $\nu'$.

**Theorem 1.** *Let $(\Omega_o, \mathcal{F}_o, \nu_s)$, $(\Omega_d, \mathcal{F}_d, \nu_{s^d})$, $(\Omega_o, \mathcal{F}_o, \nu_t)$, and $(\Omega_d, \mathcal{F}_d, \nu_{t^d})$ be four probability spaces, where $\Omega_o$ and $\Omega_d$ are disjoint and $\Omega_o \cup \Omega_d \subseteq \mathbb{R}^n$. With the results of Lemma 1, let $(\Omega_o \cup \Omega_d, \mathcal{F}_u, \nu_{s \cup s^d})$ and $(\Omega_o \cup \Omega_d, \mathcal{F}_u, \nu_{t \cup t^d})$ be two probability spaces with probability measures defined as $\nu_{s \cup s^d} = 1/2\nu_s + 1/2\nu_{s^d}$ and $\nu_{t \cup t^d} = 1/2\nu_t + 1/2\nu_{t^d}$. Specifically, when sampling from $\nu_{t \cup t^d}$, there is an equal probability of $1/2$ to sample from $\nu_t$ or $\nu_{t^d}$. Similarly, sampling from $\nu_{s \cup s^d}$ gives an equal probability of $1/2$ to draw from $\nu_s$ or $\nu_{s^d}$. Consider a classifier $C \in \mathcal{H}_1$ and an ideal classifier $C^* = \arg\min_C \varepsilon_{s \cup s^d}(C) + \varepsilon_t(C)$ satisfying the $K_L$-Lipschitz constraint, where $\mathcal{H}_1$ is a subspace of the hypothesis space $\mathcal{H}$. For every classifier $C$ in $\mathcal{H}_1$, the following inequality holds:*

$$\varepsilon_t(C) \leq 2\varepsilon_{s \cup s^d}(C) + 4K_L \cdot W_N(\nu_{s \cup s^d}, \nu_{t \cup t^d}) + \eta^*,$$

*where $\eta^* = 2\varepsilon_{s \cup s^d}(C^*) + \varepsilon_t(C^*)$ is the ideal combined error and is a sufficiently small constant.*

*Proof.* Let $Z$ be an indicator random variable that indicates whether the sample $\hat{f}_t$ is drawn from $\nu_t$ or $\nu_{t^d}$:

- $Z = 0$ if the sample is from $\nu_{t^d}$.

- $Z = 1$ if the sample is from $\nu_t$.

By the Law of Total Expectation, we have:

$$\varepsilon_{t \cup t^d}(C, C^*) = \mathbb{E}_{\hat{f}_t \sim \nu_{t \cup t^d}}[|C(\hat{f}_t) - C^*(\hat{f}_t)|]$$
$$= \mathbb{E}_{\hat{f}_t \sim \nu_{t \cup t^d}}[|C(\hat{f}_t) - C^*(\hat{f}_t)| \mid Z = 0]P(Z = 0) + \mathbb{E}_{\hat{f}_t \sim \nu_{t \cup t^d}}[|C(\hat{f}_t) - C^*(\hat{f}_t)| \mid Z = 1]P(Z = 1).$$

Substituting $P(Z = 0) = p_0$ and $P(Z = 1) = p_1$,

$$\varepsilon_{t \cup t^d}(C, C^*) = p_0 \mathbb{E}_{\hat{f}_t \sim \nu_{t \cup t^d}}[|C(\hat{f}_t) - C^*(\hat{f}_t)| \mid Z = 0] + p_1 \mathbb{E}_{\hat{f}_t \sim \nu_{t \cup t^d}}[|C(\hat{f}_t) - C^*(\hat{f}_t)| \mid Z = 1].$$

Recognize that $\mathbb{E}_{\hat{f}_t \sim \nu_{t \cup t^d}}[|C(\hat{f}_t) - C^*(\hat{f}_t)| \mid Z = 1]$ is the expectation when $\hat{f}_t$ is drawn from $\nu_t$,

$$\varepsilon_t(C, C^*) = \mathbb{E}_{\hat{f}_t \sim \nu_{t \cup t^d}}[|C(\hat{f}_t) - C^*(\hat{f}_t)| \mid Z = 1].$$

Combining these, we get:

$$\varepsilon_{t \cup t^d}(C, C^*) = p_0 \mathbb{E}_{\hat{f}_t \sim \nu_{t \cup t^d}}[|C(\hat{f}_t) - C^*(\hat{f}_t)| \mid Z = 0] + p_1 \varepsilon_t(C, C^*).$$

Then,

$$\frac{1}{p_1}\varepsilon_{t \cup t^d}(C, C^*) = \frac{p_0}{p_1} \mathbb{E}_{\hat{f}_t \sim \nu_{t \cup t^d}}[|C(\hat{f}_t) - C^*(\hat{f}_t)| \mid Z = 0] + \varepsilon_t(C, C^*).$$

Since $\frac{p_0}{p_1} \mathbb{E}_{\hat{f}_t \sim \nu_{t \cup t^d}}[|C(\hat{f}_t) - C^*(\hat{f}_t)| \mid Z = 0] \geq 0$,

$$\frac{1}{p_1}\varepsilon_{t \cup t^d}(C, C^*) \geq \varepsilon_t(C, C^*).$$

Substituting $p_1 = 1/2$, we obtain:

$$2\varepsilon_{t \cup t^d}(C, C^*) \geq \varepsilon_t(C, C^*).$$

Based on Lemma 2, we have:

$$|\varepsilon_{s \cup s^d}(C, C^*) - \varepsilon_{t \cup t^d}(C, C^*)| \leq 2K_L \cdot W_N(\nu_{s \cup s^d}, \nu_{t \cup t^d}).$$

By triangular inequality,

$$\varepsilon_t(C) \ \le \varepsilon_t(C^*) + \varepsilon_t(C^*, C),$$

$$\varepsilon_{s \cup s^d}(C, C^*) \le \varepsilon_{s \cup s^d}(C) + \varepsilon_{s \cup s^d}(C^*).$$

Then, we can derive:

$$
\begin{aligned}
\varepsilon_t(C) &\le \varepsilon_t(C^*) + \varepsilon_t(C^*, C) \\
&\le \varepsilon_t(C^*) + 2\varepsilon_{t \cup t^d}(C^*, C) \\
&= \varepsilon_t(C^*) + 2\varepsilon_{s \cup s^d}(C, C^*) + 2\varepsilon_{t \cup t^d}(C, C^*) - 2\varepsilon_{s \cup s^d}(C, C^*) \\
&\le \varepsilon_t(C^*) + 2\varepsilon_{s \cup s^d}(C, C^*) + 4K_L \cdot W_N(\nu_{s \cup s^d}, \nu_{t \cup t^d}) \\
&\le \varepsilon_t(C^*) + 2\varepsilon_{s \cup s^d}(C) + 2\varepsilon_{s \cup s^d}(C^*) + 4K_L \cdot W_N(\nu_{s \cup s^d}, \nu_{t \cup t^d}) \\
&= 2\varepsilon_{s \cup s^d}(C) + 4K_L \cdot W_N(\nu_{s \cup s^d}, \nu_{t \cup t^d}) + \eta^*.
\end{aligned}
$$

**Definition 3 ($L_1$-Transportation Cost Information Inequality). Djellout et al. (2004)** *Given $\eta > 0$, a probability measure $\nu$ on a measurable space $(\Omega, \mathcal{F})$ satisfies $T_1(\eta)$ if the inequality*

$$W_1(\nu', \nu) \le \sqrt{\frac{2}{\eta} H(\nu' | \nu)}$$

*where*

$$H(\nu' | \nu) = \int \log \frac{d\nu'}{d\nu} d\nu'$$

*holds for any probability measure $\nu'$ on $(\Omega, \mathcal{F})$, where $W_1$ represents the 1-Wasserstein distance.*

**Lemma 3. (Corollary 2.6 in Bolley & Villani (2005))** *For a probability measure $\nu$ on a measurable space $(\Omega, \mathcal{F})$, the following statements are equivalent:*

- *$\nu$ satisfies $T_1(\eta)$ inequality for some $\eta$ that can be explicitly found.*

- *$\nu$ has a square-exponential moment, i.e., there exists $\alpha > 0$ such that*

$$\int_\Omega \exp(\alpha d(x, y)^2) \, d\nu(x) \text{ is finite}$$

*for any $y \in \Omega$. Here, $d$ is a measurable distance over $\Omega$.*

**Lemma 4.** *Let $(\Omega_1, \mathcal{F}_1, \nu_1)$ and $(\Omega_2, \mathcal{F}_2, \nu_2)$ be two probability spaces, where $\Omega_1$ and $\Omega_2$ are disjoint. Let $p_1, p_2 \in [0, 1]$ be constants such that $p_1 + p_2 = 1$. Define a new measure $\nu_3$ on a measurable space $(\Omega_3, \mathcal{F}_3)$, where $\Omega_3 = \Omega_1 \cup \Omega_2$:*

$$\nu_3(A) := p_1 \nu_1(A \cap \Omega_1) + p_2 \nu_2(A \cap \Omega_2), \quad \forall A \in \mathcal{F}_3.$$

*Assume that $\nu_1$ and $\nu_2$ admit square-exponential moments: for some constants $\alpha_1, \alpha_2 > 0$, and for all $y_1 \in \Omega_1$, $y_2 \in \Omega_2$,*

$$\int_{\Omega_1} \exp\left(\alpha_1 d_1(x, y_1)^2\right) d\nu_1(x) < \infty, \quad \int_{\Omega_2} \exp\left(\alpha_2 d_2(x, y_2)^2\right) d\nu_2(x) < \infty,$$

*where $d_1 : \Omega_1 \times \Omega_1 \to [0, \infty)$ and $d_2 : \Omega_2 \times \Omega_2 \to [0, \infty)$ are distance functions. Then, $\nu_3$ is a probability measure on $(\Omega_3, \mathcal{F}_3)$, and there exists a distance function $d : \Omega_3 \times \Omega_3 \to [0, \infty)$ and a constant $\alpha \in (0, \min\{\alpha_1, \alpha_2\})$ such that $\nu_3$ admits a square-exponential moment: for all $y \in \Omega_3$,*

$$\int_{\Omega_3} \exp\left(\alpha d(x, y)^2\right) d\nu_3(x) < \infty.$$

*Proof.* First, we define $d : \Omega_3 \times \Omega_3 \to [0, \infty)$:

$$d(x, y) = \begin{cases} d_1(x, y) & \text{if } x, y \in \Omega_1 \\ d_2(x, y) & \text{if } x, y \in \Omega_2 \\ C & \text{if } x \in \Omega_1, y \in \Omega_2 \text{ (or vice versa)} \end{cases}$$

where $C$ is a finite constant chosen to ensure $d$ is a metric on $\Omega_3$. $d$ can be expressed as:

$$d(x, y) = d_1(x, y)\mathbf{1}_{x,y \in \Omega_1} + d_2(x, y)\mathbf{1}_{x,y \in \Omega_2} + C\mathbf{1}_{x \in \Omega_1, y \in \Omega_2 \text{ or } x \in \Omega_2, y \in \Omega_1},$$

where $\mathbf{1}$ is the indicator function. Accodring to Theorem 1.9 (d) in Rudin (1987), the indicator functions are measurable because $\Omega_1 \times \Omega_1$, $\Omega_2 \times \Omega_2$, and $(\Omega_1 \times \Omega_2) \cup (\Omega_2 \times \Omega_1)$ are all measurable sets in $\Omega_3 \times \Omega_3$. $d_1$ and $d_2$ are also measurable functions by assumption. Therefore, $d$ is a sum of products of measurable functions. Hence, $d$ is measurable. For any $y$ in $\Omega_1$,

$$\int_{\Omega_3} \exp(\alpha d(x, y)^2)\, d\nu_3(x)$$

$$= p_1 \int_{\Omega_1} \exp(\alpha d_1(x, y)^2)\, d\nu_1(x) + p_2 \int_{\Omega_2} \exp(\alpha d(x, y)^2)\, d\nu_2(x)$$

$$\leq p_1 \int_{\Omega_1} \exp(\alpha_1 d_1(x, y)^2)\, d\nu_1(x) + p_2 \int_{\Omega_2} \exp(\alpha C^2)\, d\nu_2(x)$$

$$= p_1 \int_{\Omega_1} \exp(\alpha_1 d_1(x, y)^2)\, d\nu_1(x) + p_2 \exp(\alpha C^2) < \infty.$$

For any $y$ in $\Omega_2$,

$$\int_{\Omega_3} \exp(\alpha d(x, y)^2)\, d\nu_3(x)$$

$$= p_1 \int_{\Omega_1} \exp(\alpha d(x, y)^2)\, d\nu_1(x) + p_2 \int_{\Omega_2} \exp(\alpha d_2(x, y)^2)\, d\nu_2(x)$$

$$\leq p_1 \int_{\Omega_1} \exp(\alpha C^2)\, d\nu_1(x) + p_2 \int_{\Omega_2} \exp(\alpha_2 d_2(x, y)^2)\, d\nu_2(x)$$

$$= p_1 \exp(\alpha C^2) + p_2 \int_{\Omega_2} \exp(\alpha_2 d_2(x, y)^2)\, d\nu_2(x) < \infty.$$

This proves that $\nu_3$ has a square-exponential moment for some $0 < \alpha < \min(\alpha_1, \alpha_2)$.

**Lemma 5.  (Theorem 1.1 of  Bolley et al. (2007))** *Let $\nu$ be a probability measure on $(\Omega, \mathcal{F})$ where $\Omega \subseteq \mathbb{R}^n$. $\nu$ satisfies a $T_1(\eta)$ inequality. Let $\hat{\nu} = \frac{1}{N} \sum_{i=1}^N \delta_{f^i}$ be its associated empirical measure defined on a sample set $\{f^i\}_{i=1}^N$ of size $N$ drawn i.i.d from $\nu$. Then for any $n' > n$ and $\eta' < \eta$, there exists some constant $N_0$ depending on $n'$ and some square-exponential moment of $\nu$ such that for any $\epsilon > 0$ and $N \geq N_0 \max(\epsilon^{-(n'+2)}, 1)$, the following holds:*

$$\mathbb{P}[W_N(\nu, \hat{\nu}) > \epsilon] \leq \exp\left(-\frac{\eta'}{2} N\epsilon^2\right).$$

**Theorem 2.**  *Under the assumption of Theorem 1, $\Omega_o$ and $\Omega_d$ are disjoint and $\Omega_o \cup \Omega_d \subseteq \mathbb{R}^n$. Let $(\Omega_o \cup \Omega_d, \mathcal{F}_u, \nu_{s \cup s^d})$ and $(\Omega_o \cup \Omega_d, \mathcal{F}_u, \nu_{t \cup t^d})$ be two probability spaces with $\nu_{s \cup s^d} = 1/2\nu_s + 1/2\nu_{s^d}$ and $\nu_{t \cup t^d} = 1/2\nu_t + 1/2\nu_{t^d}$, where $\nu_s, \nu_{s^d}, \nu_t, \nu_{t^d}$ each has a square-exponential moment. From Lemma 3 and 4, $\nu_{s \cup s^d}$ satsifies $T_1(\eta_s)$ for some $\eta_s$ and $\nu_{t \cup t^d}$ satsifies $T_1(\eta_t)$ for some $\eta_t$. Let $\{\hat{f}_s^i\}_{i=1}^{N_s}$ and $\{\hat{f}_t^i\}_{i=1}^{N_t}$ be two sample sets of size $N_s$ and $N_t$ drawn i.i.d from $\nu_{s \cup s^d}$ and $\nu_{t \cup t^d}$, respectively. $\hat{\nu}_{s \cup s^d} = \frac{1}{N_s} \sum_{i=1}^{N_s} \delta_{\hat{f}_s^i}$ and $\hat{\nu}_{t \cup t^d} = \frac{1}{N_t} \sum_{i=1}^{N_t} \delta_{\hat{f}_t^i}$ are associated empirical probability measures. Then, for any $n' > n$ and $\eta' < \min(\eta_s, \eta_t)$,*

there exists a constant $N_0$ depending on $n'$ such that for any $\delta > 0$ and $\min(N_s, N_t) \geq N_0 \max(\delta^{-(n'+2)}, 1)$, with probability at least $1 - \delta$, the following holds for all $C$:

$$\varepsilon_t(C) \leq 2\varepsilon_{s \cup s^d}(C) + 4K_L \cdot W_N(\hat{\nu}_{s \cup s^d}, \hat{\nu}_{t \cup t^d}) + \eta^* + 4K_L \cdot \sqrt{\frac{2}{\eta'} \log \frac{1}{\delta}} \left( \sqrt{\frac{1}{N_s}} + \sqrt{\frac{1}{N_t}} \right), \qquad (17)$$

where $\eta^* = 2\varepsilon_{s \cup s^d}(C^*) + \varepsilon_t(C^*)$ is the ideal combined error and is a sufficiently small constant.

*Proof.* Based on Theorem 1,

$$\varepsilon_t(C) \leq 2\varepsilon_{s \cup s^d}(C) + 4K_L \cdot W_N(\nu_{s \cup s^d}, \nu_{t \cup t^d}) + \eta^*.$$

As a part of a broader class of Wasserstein distances, $W_N$ satisfies the axioms of a distance Villani et al. (2009). Hence, $W_N$ satisfies the triangle inequality:

$$\varepsilon_t(C) \leq 2\varepsilon_{s \cup s^d}(C) + 4K_L \cdot W_N(\nu_{s \cup s^d}, \hat{\nu}_{s \cup s^d}) + 4K_L \cdot W_N(\hat{\nu}_{s \cup s^d}, \nu_{t \cup t^d}) + \eta^*$$
$$\leq 2\varepsilon_{s \cup s^d}(C) + 4K_L \cdot W_N(\nu_{s \cup s^d}, \hat{\nu}_{s \cup s^d}) + 4K_L \cdot W_N(\hat{\nu}_{s \cup s^d}, \hat{\nu}_{t \cup t^d}) + 4K_L \cdot W_N(\hat{\nu}_{t \cup t^d}, \nu_{t \cup t^d}) + \eta^*.$$

From Lemma 5,

$$W_N(\nu_{s \cup s^d}, \hat{\nu}_{s \cup s^d}) \leq \sqrt{\frac{2}{\eta'} \log \left( \frac{1}{\delta} \right)} \cdot \sqrt{\frac{1}{N_s}},$$

$$W_N(\nu_{t \cup t^d}, \hat{\nu}_{t \cup t^d}) \leq \sqrt{\frac{2}{\eta'} \log \left( \frac{1}{\delta} \right)} \cdot \sqrt{\frac{1}{N_t}}.$$

$W_N$ belongs to the family of 1-Wasserstein distance. By the symmetry property of distance,

$$W_N(\hat{\nu}_{s \cup s^d}, \nu_{s \cup s^d}) = W_N(\nu_{s \cup s^d}, \hat{\nu}_{s \cup s^d}) \leq \sqrt{\frac{2}{\eta'} \log \left( \frac{1}{\delta} \right)} \cdot \sqrt{\frac{1}{N_s}},$$

$$W_N(\hat{\nu}_{t \cup t^d}, \nu_{t \cup t^d}) = W_N(\nu_{t \cup t^d}, \hat{\nu}_{t \cup t^d}) \leq \sqrt{\frac{2}{\eta'} \log \left( \frac{1}{\delta} \right)} \cdot \sqrt{\frac{1}{N_t}}.$$

Substituting back, we have:

$$\varepsilon_t(C) \leq 2\varepsilon_{s \cup s^d}(C) + 4K_L \cdot W_N(\hat{\nu}_{s \cup s^d}, \hat{\nu}_{t \cup t^d}) + \eta^* + 4K_L \cdot \sqrt{\frac{2}{\eta'} \log \frac{1}{\delta}} \left( \sqrt{\frac{1}{N_s}} + \sqrt{\frac{1}{N_t}} \right).$$

## A.2 Implementation Details

Our code is based on the open-source implementation of the DefRec+PCM. We trained our three CDND models with seeds {1, 2, 3} on A100 GPUs. For the PointSegDA dataset, we fixed the learning rate to be 0.001 and conducted a grid search to optimize the hyperparameters $\alpha$, $\gamma$, $\beta_1$, and $\beta_2$ for each task. The specific hyperparameter values can be found in Table 5. Similarly, for the PointDA dataset, the hyperparameters are listed in Table 4. Training on the PointDA dataset takes approximately 10 hours, and around 8 hours required for source-only training without any adaptation method. This results in a high computational cost for hyperparameter tuning. Therefore, we do not tune the hyperparameters extensively. Similarly, for GAST and ImplicitPCDA, we use the hyperparameters provided in their open-source code for the PointDA dataset.

However, GAST and ImplicitPCDA have not been tested on the PointSegDA dataset before. When implementing GAST, we conduct a grid search on the PointSegDA dataset, exploring values of 0.1, 0.2, 0.5, and 1.0 for both $\mathcal{L}_{\text{rot}}$ and $\mathcal{L}_{\text{loc}}$. For ImplicitPCDA, we perform a grid search on the PointSegDA dataset, considering values of 0.1, 0.2, 0.5, and 1.0 for $\mathcal{L}_M$. Please refer to the original papers for the definitions of $\mathcal{L}_{\text{rot}}$, $\mathcal{L}_{\text{loc}}$, and $\mathcal{L}_M$.

For our method, we set the number of regions to be deformed to 5 for PointDA and 10 for PointSegDA. Each region contains 55 points. The total number of regions is set to 20 for the PointDA dataset and 40 for the PointSegDA dataset. For DefRec+PCM and CurvRec+PCM, we follow the hyperparameter settings used in the open-source implementation of DefRec and PCM (PointMixup).

| Hyperparameter | Values |
|---|---|
| Learning Rate | 0.001, 0.0001 (S$^+$M, MS) |
| $\alpha$ | 0.5 |
| $\gamma$ | 0.5 |
| $\beta_1$ | [0.0, 1.0] |
| $\beta_2$ | 0.2 |
| # of Epochs | 200 |

Table 4: Hyperparameters for PointDA.

| Hyperparameter | Values |
|---|---|
| Learning Rate | 0.001 |
| $\alpha$ | 1.0 |
| $\gamma$ | [0.05, 0.1, 0.2, 0.5, 1.0] |
| $\beta_1$ | [0.0, 0.05, 0.1, 0.2, 0.5, 1.0] |
| $\beta_2$ | [0.0, 0.2] |
| # of Epochs | 150 |

Table 5: Hyperparameters for PointSegDA.

**Challenges of Applying SPST with mIoU**  The mIoU metric is defined as:

$$\text{mIoU} = \frac{1}{M} \sum_{m=1}^{M} \frac{TP_m}{TP_m + FP_m + FN_m}$$

$$\text{where:} \begin{cases} M = \text{number of classes} \\ TP_m = \text{true positive for class } m \\ FP_m = \text{false positive for class } m \\ FN_m = \text{false negative for class } m \end{cases}$$

SPST typically relies on ranking training samples by difficulty and gradually incorporating harder examples into training. The training samples for point cloud segmentation tasks are points in point clouds. However, mIoU is a global metric that evaluates performance across an entire point cloud, making it challenging to assign difficulty scores to individual points in a point cloud. The mechanism of SPST mismatches the per-point cloud, rather than per-point, evaluation criterion of mIoU.

## A.3   Additional Results

We also added visualization comparisons on ModelNet to ScanNet task in Figure 3, featuring three methods: GAST and DefRec (which achieved second and third-best scores on the ModelNet to ScanNet task) and PCFEA (the most recently published method). Our approach achieves the most distinct and well-separated class clusters (represented by distinct colors) on the target domain test set, while PCFEA and GAST show considerable class mixing, and DefRec shows a tight clustering of some groups, lacking a clear separation.

We present segmentation visualization results in Figure 4 for the PointSegDA dataset. Different segmentation parts are highlighted using different colors. We selected three domains—MA, MF, and SA—to demonstrate the effectiveness of our CDND method, which achieves state-of-the-art performance with a significant margin over the second-best method. In the figure, the last row labeled "GT" shows the ground truth. Our CDND results are shown in the third row. Although CDND achieves 2% improvement in mIoU compared to RS Sauder & Sievers (2019), we include this comparison to highlight the qualitative differences in segmentation. The "Unsupervised" row represents results without any domain adaptation technique. From Figure 4, it is evident that CDND produces segmentation results most closely aligned with the ground truth, outperforming both RS and the unsupervised baseline. Additionally, It is worth noting that other methods also report small increases in mIoU margins on PointSegDA dataset.

We also conduct experiments to evaluate the effects of different Gaussian variances and the total number of regions in CurvRec. These experiments are performed on the FAUST to ADOBE task of the PointSegDA dataset, and the results are presented in Table 6. Specifically, we vary the total number of regions, $k \in \{10, 25, 40, 55, 70, 85, 100\}$, and the Gaussian variance, $\sigma^2 \in \{0.0001, 0.001, 0.01, 0.1\}$. The number of points per region, $N_k$, is calculated as $\lfloor \frac{2048}{k} \rfloor + 4$, where 2048 is the number of points per sample. The additional 4 points ensure that $k \times N_k \geq 2048$. This condition is important because we want to maximize coverage of the input point cloud—ideally assigning all 2048 points to regions. Without this constraint, $\lfloor \frac{2048}{k} \rfloor$ could be less than 2048, leaving some points uncovered. By slightly increasing the number of points per region, we ensure complete or near-complete coverage. As shown in the table, the optimal performance is achieved when the

variance is set to 0.001 and the number of regions is 40, which aligns with our experimental settings. The results also indicate that model performance is relatively insensitive to the number of regions but is more sensitive to the choice of variance. In particular, large variances (e.g., 0.1) lead to more uniform deformations and may negatively affect performance.

| $\sigma^2 \setminus (k, N_k)$ | (10, 208) | (25, 85) | **(40, 55)** | (55, 41) | (70, 33) | (85, 28) | (100, 24) |
|---|---|---|---|---|---|---|---|
| 0.0001 | $80.5_{\pm0.9}$ | $80.3_{\pm2.8}$ | $79.8_{\pm1.9}$ | $79.6_{\pm1.2}$ | $79.4_{\pm1.0}$ | $80.6_{\pm1.7}$ | $80.1_{\pm1.7}$ |
| 0.001 | $80.8_{\pm1.2}$ | $80.7_{\pm1.8}$ | $\mathbf{81.5_{\pm2.0}}$ | $80.9_{\pm1.7}$ | $79.2_{\pm2.3}$ | $78.9_{\pm2.1}$ | $79.1_{\pm2.0}$ |
| 0.01 | $78.6_{\pm2.9}$ | $80.3_{\pm2.3}$ | $81.0_{\pm1.9}$ | $79.3_{\pm4.0}$ | $79.9_{\pm2.3}$ | $79.6_{\pm3.0}$ | $77.5_{\pm4.0}$ |
| 0.1 | $69.0_{\pm2.8}$ | $68.3_{\pm3.6}$ | $68.2_{\pm5.6}$ | $67.2_{\pm4.4}$ | $67.9_{\pm4.9}$ | $69.9_{\pm4.5}$ | $67.0_{\pm5.2}$ |

Table 6: Results for varying $\sigma^2$, number of regions $k$, and points per region $N_k$.

Another important hyperparameter is the number of regions selected for deformation. In Table 7, we present results for the FAUST to ADOBE task of the PointSegDA dataset, where the total number of regions is fixed at $k = 40$ and the Gaussian variance is set to $\sigma^2 = 0.001$. We vary the number of deformed regions $N$ (expressed as the ratio $N/k$) to observe its effect on performance. The results show that performance is not sensitive to the number of deformed regions. The optimal performance is achieved when the ratio is set to 0.25, which aligns with our experimental settings.

| Ratio ($N/k$) | 0.10 | **0.25** | 0.30 | 0.45 |
|---|---|---|---|---|
| FA | $80.3_{\pm3.0}$ | $\mathbf{81.5_{\pm2.0}}$ | $81.1_{\pm2.3}$ | $79.5_{\pm3.3}$ |

Table 7: Results for varying the number of deformed regions $N$, expressed as the ratio $N/k$.

In the theoretical section, we note that the results can be generalized to arbitrary sampling ratios by adjusting the mixture weights in the underlying probability measures. While it is theoretically challenging to prove that a specific sampling ratio between original and deformed samples (e.g., 1:1, 2:3, etc.) is universally optimal, our experimental results demonstrate that our chosen setting yields high performance. We conduct experiments on the FAUST to ABODE task of the PointSegDA dataset. As shown in Table 8, we achieve the highest mIoU with the 1:1 and 4:3 ratios. This supports the intuition that a balanced and sufficiently diverse mix of original and deformed samples helps the model generalize more effectively across domains. It is also worth noting that our method is not highly sensitive to changes in the sampling ratio.

We also conducted additional experiments to analyze the impact of hyperparameters on curvature estimation. Curvature estimation depends on clean local geometry and can be influenced by real-world scanning artifacts. To evaluate, we report classification accuracy on the ModelNet to ShapeNet task from the PointDA dataset, and segmentation performance (mIoU) on the FAUST to ADOBE task from the PointSegDA dataset. Table 9 presents the results of varying the number of neighbors used in curvature computation. The performance remains relatively stable (though the segmentation task is more sensitive) across different neighborhood sizes, indicating that curvature estimation is not significantly sensitive to this parameter. Notably, our experiment

| Ratio | 1:1 | 1:2 | 1:3 | 1:4 | 2:1 | 2:3 | 2:4 | 3:1 | 3:2 | 3:4 | 4:1 | 4:2 | 4:3 |
|---|---|---|---|---|---|---|---|---|---|---|---|---|---|
| FA | $81.5_{\pm2.0}$ | $80.1_{\pm1.1}$ | $79.3_{\pm1.1}$ | $79.5_{\pm3.9}$ | $79.5_{\pm0.9}$ | $79.3_{\pm1.1}$ | $79.5_{\pm3.9}$ | $76.7_{\pm2.8}$ | $79.9_{\pm2.0}$ | $79.5_{\pm3.9}$ | $77.2_{\pm3.5}$ | $77.1_{\pm3.0}$ | $\mathbf{81.7_{\pm1.2}}$ |

Table 8: Results for various original:deformed sample ratios.

| Neighborhood | 10 | 20 | 40 | CDND (32) |
|---|---|---|---|---|
| MS (Accuracy) | $\mathbf{84.1_{\pm0.3}}$ | $83.7_{\pm0.4}$ | $83.6_{\pm0.4}$ | $\mathbf{84.1_{\pm0.3}}$ |
| FA (mIoU) | $77.7_{\pm2.9}$ | $76.7_{\pm3.5}$ | $78.4_{\pm2.6}$ | $\mathbf{81.5_{\pm2.0}}$ |

Table 9: Results for various neighborhoods.

| Gaussian Variance | 0.0001 | 0.01 | 0.1 | Clean |
|---|---|---|---|---|
| MS (Accuracy) | $83.2_{\pm0.5}$ | $83.5_{\pm0.4}$ | $82.7_{\pm1.5}$ | $\mathbf{83.6_{\pm0.4}}$ |
| FA (mIoU) | $79.9_{\pm1.3}$ | $75.8_{\pm5.1}$ | $56.4_{\pm1.9}$ | $\mathbf{78.4_{\pm2.6}}$ |

Table 10: Results for adding Gaussian noise.

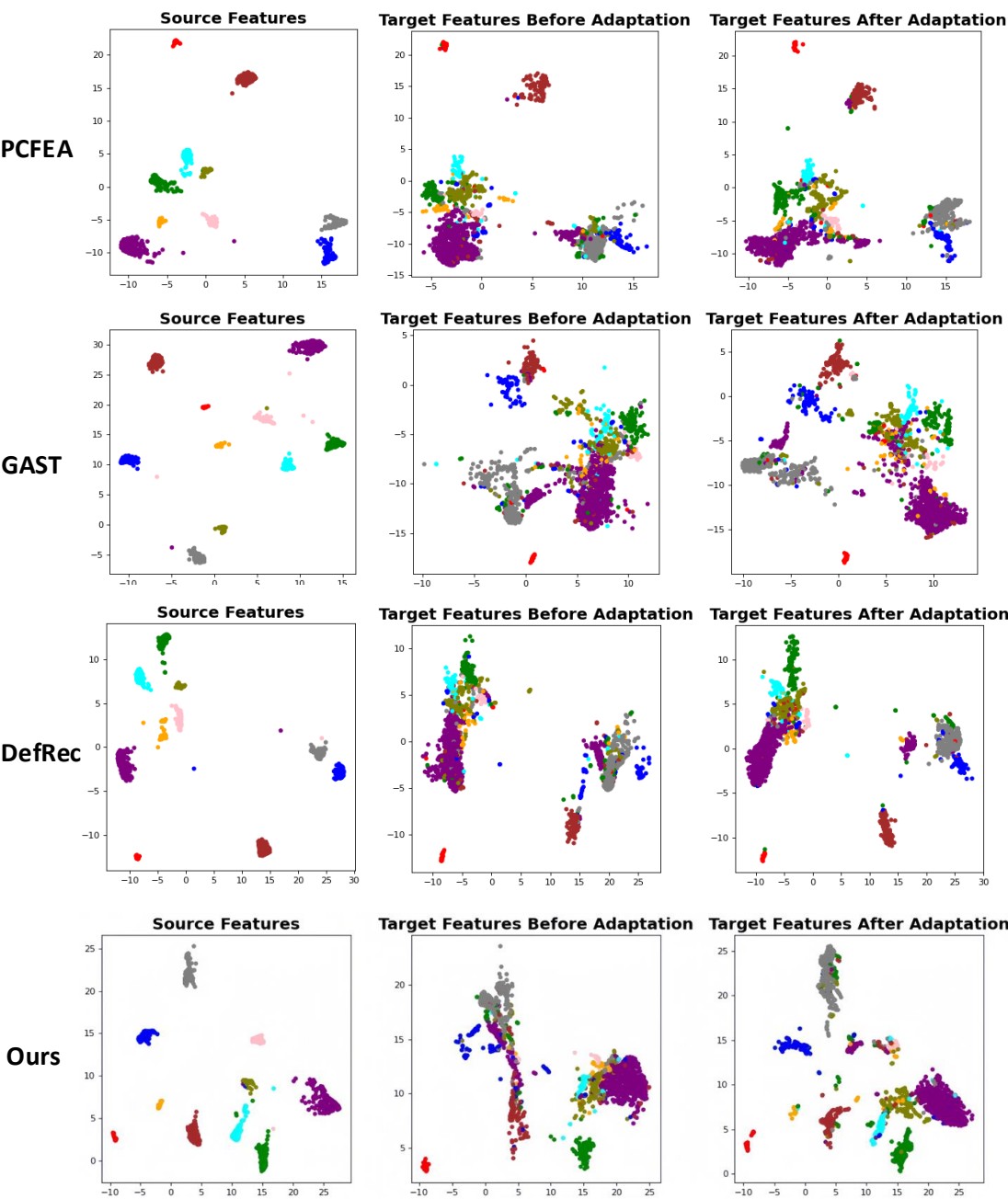

Figure 3: Umap results of other baselines on PointDA-10 dataset.

setting of 32 neighbors (labeled as CDND) yields the best overall performance for both tasks. Regarding the number of principal components, we do not arbitrarily choose how many to use — the number is determined by the dimensionality of the input point cloud data. In our case, the input points lie in 3D space, so the covariance matrix is $3 \times 3$, yielding three eigenvalues and corresponding eigenvectors.

In Table 10, we introduce Gaussian noise to the point coordinates to evaluate the robustness of curvature estimation. For these experiments, the number of neighbors is fixed at 40. The classification task (ModelNet to ShapeNet) demonstrates robustness to noise, as the accuracy remains relatively consistent even with increasing noise variance. However, the segmentation task (FAUST to ADOBE) is more sensitive—its

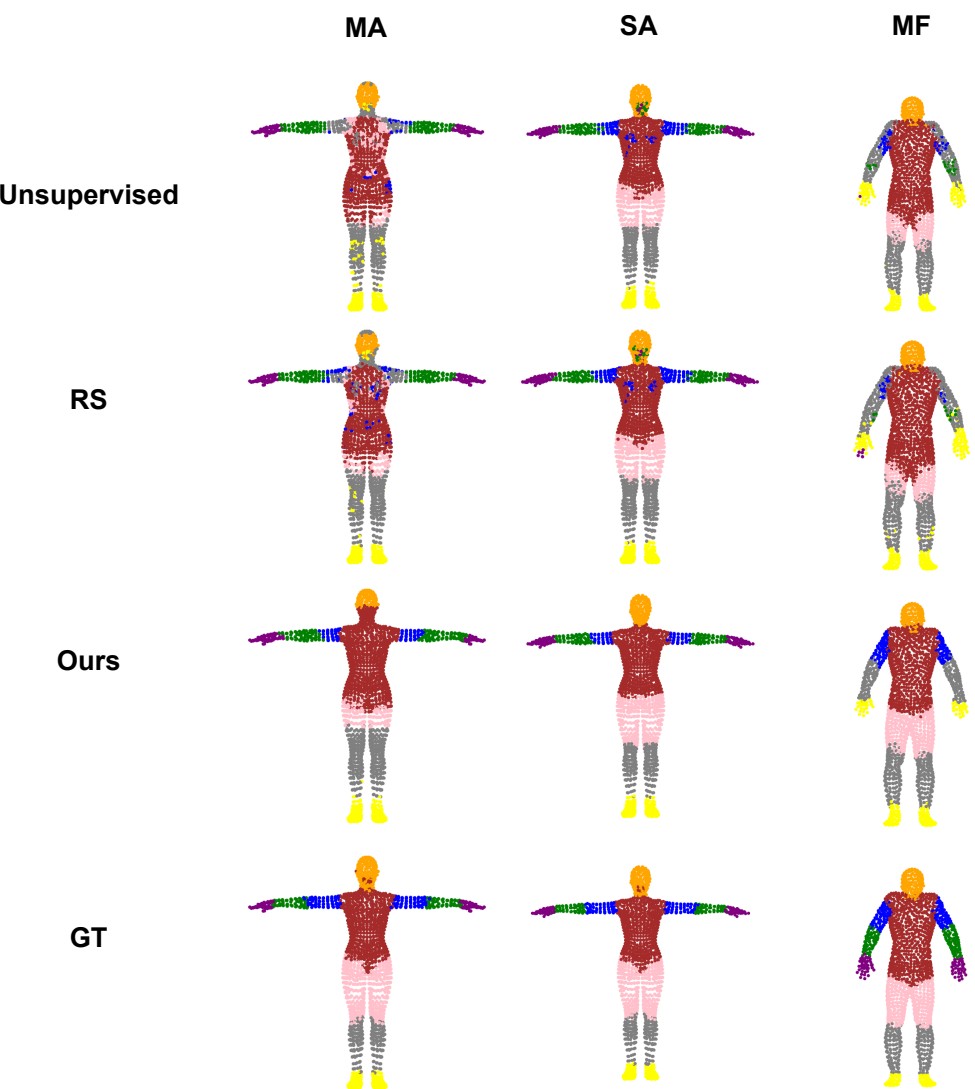

Figure 4: Segmentation visualization results of PointSegDA dataset.

performance degrades significantly with higher noise levels. This increased sensitivity in segmentation arises because segmentation tasks require more precise, point-level geometric details to assign accurate labels to each point. Noise distorts these fine-grained features, making it harder for the model to distinguish between regions, especially in complex structures. In contrast, classification tasks benefit from more global shape features and can tolerate localized noise to a greater extent.

| Method | Unsupervised | DANN | PointDAN | RS | DefRec + PCM | PCFEA | GAST | ImplicitPCDA | CDND |
|---|---|---|---|---|---|---|---|---|---|
| Time (hours) | 8.0 | 9.0 | 9.2 | 35.9 | 10.4 | 8.3 | 18.9 | 9.8 | 10.4 |

Table 11: Training time (in hours) of different methods.

Table 11 presents the training time (in hours) for various baseline methods along with our proposed approach, CDND. As observed, our method does not introduce a significant computational overhead compared to the "Unsupervised" baseline (without any domain adaptation method), requiring extra 2.4 additional hours. This demonstrates a decent efficiency of CDND, especially when contrasted with methods like RS and GAST,

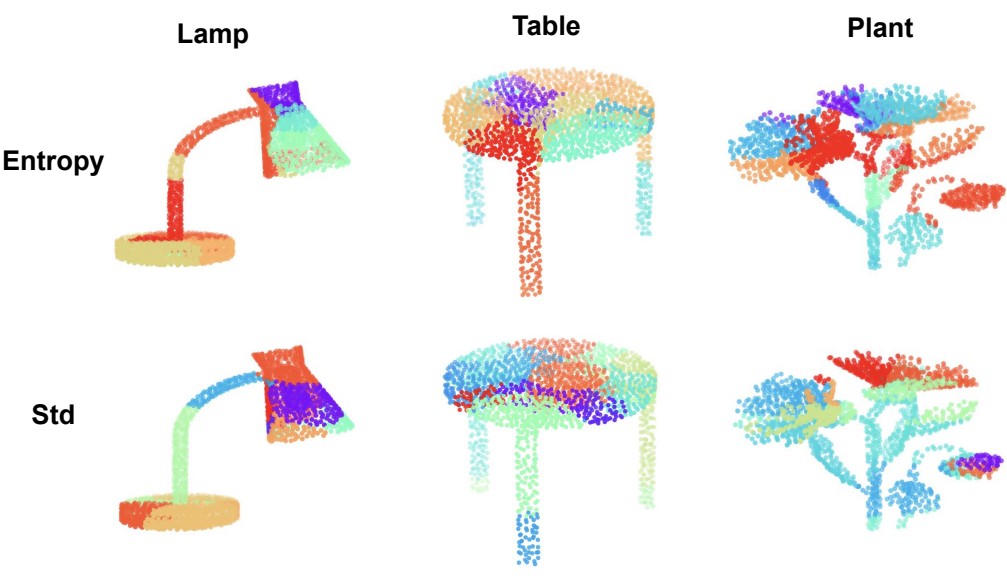

Figure 5: Visualization comparisons between curvature diversity metrics (entropy/standard deviation).

which have significantly higher training costs. Furthermore, CDND's training time is still comparable to DANN, PointDAN, and DefRec+PCM. Overall, CDND is practically viable without significantly sacrificing efficiency.

To better illustrate the advantages of using entropy to evaluate curvature diversity, we provide visual comparisons between entropy- and standard deviation-based curvature diversity measures on ModelNet-10 dataset in Figure 5. In these visualizations, warmer colors indicate regions with higher semantic richness and curvature diversity. We observe that the entropy-based measure more effectively highlights structurally and semantically important regions, such as the joints between object components. For example, in the lamp, both the connection between the cap and the supporting rod, as well as the joint between the rod and the base, are clearly highlighted in red by entropy, indicating high curvature diversity. Similarly, in the table, the interfaces between the tabletop and its supporting legs are also captured with warm colors, emphasizing their semantic importance. In contrast, the standard deviation measure fails to consistently highlight these joint regions. Furthermore, in the plant example, most of the leaves and flower structures—which are semantically rich and geometrically intricate—are highlighted by entropy with warm tones. However, these same regions are mostly shown in cooler colors (e.g., blue/green) under the standard deviation measure, indicating a failure to recognize their informative structure. Overall, these visualizations demonstrate that entropy provides a reliable and semantically aligned indicator of curvature diversity.

Additionally, in Table 2, CurvRec(En)-High performs worse than CurvRec(S)-High. This result also supports the superiority of entropy in identifying semantically rich regions. Entropy is more effective than standard deviation in capturing areas with high curvature diversity and semantic importance. In the CurvRec(En)-High experiment, we intentionally deform the parts identified as having the highest curvature diversity—i.e., those that are semantically rich. Since entropy more accurately identifies these meaningful regions, deforming them results in a greater loss of critical geometric information, which negatively impacts model performance. On the other hand, CurvRec(S)-High, which uses standard deviation to select parts for deformation, may incorrectly classify geometrically simple or less informative regions as "important." As a result, fewer truly semantically rich regions are deformed, and more informative structures are preserved. This allows the model to continue learning from these preserved rich regions, leading to better performance compared to

CurvRec(En)-High. Therefore, the fact that CurvRec(En)-High leads to a larger performance drop actually reinforces our claim: entropy is more accurate in identifying semantically rich parts, and its deformation has a more significant impact on the model.

