# OpenReview forum: "Curvature Diversity-Driven Deformation and Domain Alignment for Point Cloud"
_TMLR — Accepted by TMLR_

### Review · Reviewer_1AYa · 2025-06-27

**Summary Of Contributions:**

The paper proposes CDND, a two–part UDA framework that (i) selects low-entropy, low-curvature “unimportant” regions for Gaussian deformation (CurvRec) and (ii) aligns the joint distribution of original + deformed features across domains via a Deformation-based Nuclear-norm Wasserstein Discrepancy (D-NWD).  Experiments on PointDA-10 (classification) and PointSegDA (segmentation) show consistent improvements over previous methods.

**Audience:**

Yes

**Broader Impact Concerns:**

The authors do not provide a detailed broader impact statement. Since this paper focuses on domain adaptation for 3D point cloud data in tasks like classification and segmentation, its techniques could be deployed in safety‐critical systems (e.g., autonomous driving, robotics navigation) where failure in domain generalization may cause unintended consequences.

**Claims And Evidence:**

Yes

**Requested Changes:**

1.	Equation for $I_e$ is unclear.
The explanation for I_e suggests “sum around both indices i and j,” but the equation instead shows a sum over off‐diagonal entries. Please rephrase to clearly state that $I_e = \sum_{i \neq j} Z_{ij}$
without the confusing “around both” wording.

2.	Repeated symbols.
Several symbols are reused in ways that create confusion. For instance, K is first introduced as the number of eigenvalues, but then also appears as a Lipschitz constant. Likewise, C denotes a classifier, but later \|C\|F is used, which is ambiguous — is this the classifier weight or its output matrix? Please use distinct symbols for these, such as K\lambda for the eigenvalue count, and something like \mathcal{C} or P for the classifier output.

3.	Undefined distributions $\nu_s$ and $\nu_t$.
These are introduced in Eq. (10), but never precisely defined. I can only guess that they are uniform empirical measures on batches, but this should be clearly stated in the text. Precise definitions are essential since the later theoretical results depend on the support of these measures.

4.	Motivation for combining D-NWD with PointMixup is not strong.
D-NWD is used to align the domain distributions, while PointMixup is a data augmentation. Since both involve deformations, the paper should explain more explicitly why PointMixup alone is insufficient, and why an extra D-NWD term is necessary. At the moment, this feels ad hoc. An ablation or a stronger theoretical argument would improve confidence.

5.	Missing hyperparameter settings. These should be clearly listed, perhaps in an appendix.

Several details are missing for reproducibility. In particular:

\begin{enumerate}
\item how many curvature regions k are actually selected

\item how many center points are sampled for D-NWD

\item the values or search ranges of $\gamma$, $alpha$ for mixup, the D-NWD loss weight

\item optimizer parameters such as learning rate, training epochs
\end{enumerate}

**Strengths And Weaknesses:**

Strengths:
1. Curvature-driven region selection is intuitive and empirically effective; entropy-based diversity outperforms standard-deviation baselines by ~3 points mAcc (Table 2) .

2. Theoretical analysis: Theorem 1-2 extend NWD bounds to the mixed distribution $\nu_{s\cup sd},\nu_{t\cup td} $.

3. Comprehensive experiments on two benchmarks; CDND improves Avg Acc from 68.9 (GAST + SPST) to 73.3 (Table 1) and Avg mIoU from 57.9 (RS) to 59.9 (Table 3) .

Weaknesses:

1. Writing clarity: Several overloaded symbols (see §2-B).

2. Complexity analysis: CurvRec adds entropy computation each epoch.  Authors claim “negligible”; please report actual overhead (ms per batch) and memory cost.

3. Ablation on deformation N: How sensitive is performance to the number of regions N and Gaussian variance? Some parameters are not set

4. Segmentation results: Improvement on PointSegDA is smaller (≤2 mIoU).  Provide qualitative visualizations or per-part IoU to show where CurvRec helps.

---

> ### Author Response · Authors · 2025-07-02
> **Rebuttal**
>
> **Weaknesses:**
>
> **1. Writing clarity: Several overloaded symbols (see §2-B).**
>
> Thank you for your valuable comments. We have revised our manuscript and the revised version is uploaded.
>
> **2. Complexity analysis: CurvRec adds entropy computation each epoch. Authors claim “negligible”; please report actual overhead (ms per batch) and memory cost.**
>
> To evaluate the computational cost of the curvature-based deformation method during training, we measure its runtime and extra GPU memory usage for each batch over 20 epochs. Here, “extra” refers to memory usage beyond that required for model parameters and input data. We then compute the average and standard deviation of both per-batch runtimes and memory usage.
>
> Avg time per batch: 117 $\pm$ 0.095 ms
>
> Avg memory per batch: 15.2 $\pm$ 0.27 MB
>
> **3. Ablation on deformation N: How sensitive is performance to the number of regions N and Gaussian variance? Some parameters are not set**
>
> Thank you for your insightful comments. In Table 7 in the revised manuscript (or below), we present results for the FAUST to ADOBE task of the PointSegDA dataset, where the total number of regions is fixed at $k = 40$ and the Gaussian variance is set to $\sigma^2 = 0.001$. We vary the number of deformed regions $N$ (expressed as the ratio $N/k$) to observe its effect on performance. The results show that performance is not sensitive to the number of deformed regions. As shown in the table, the optimal performance is achieved when the ratio is set to 0.25, which aligns with our experimental settings.
>
>
>
> | Ratio ($N/k$) | 0.10         | **0.25**       | 0.30         | 0.45         |
> |---------------|--------------|----------------|--------------|--------------|
> | FA         | 80.3±3.0     | **81.5±2.0**    | 81.0±2.3     | 79.5±3.3    |
>
>
>
> Also, as suggested, we add experiments to evaluate the effects of different Gaussian variances and the total number of regions. These experiments are performed on the FAUST to ADOBE task of the PointSegDA dataset as well, and the results are presented in Table 6 in the revised manuscript (or below). Specifically, we vary the total number of regions, $k$ in {10, 25, 40, 55, 70, 85, 100}, and the Gaussian variance, $\sigma^2$ in {0.0001, 0.001, 0.01, 0.1}. The number of points per region, $N_k = \left\lfloor \frac{2048}{k} \right\rfloor + 4$, where 2048 is the number of points per sample. The additional 4 points ensure that $k \times N_k \geq 2048$. Without this constraint,  $\left\lfloor \frac{2048}{k} \right\rfloor$ could be less than 2048, leaving some points uncovered. By slightly increasing the number of points per region, we ensure complete or near-complete coverage. As shown in the table, the optimal performance is achieved with our experimental settings. The results also indicate that model performance is relatively insensitive to the number of regions but is more sensitive to the choice of variance. In particular, large variances (e.g., 0.1) lead to more uniform deformations and may negatively affect performance.
>
>
> | $\sigma^2$ \ $(k, N_k)$ | (10, 208)      | (25, 85)      | **(40, 55)**       | (55, 41)      | (70, 33)      | (85, 28)      | (100, 24)     |
> |-------------------------|----------------|----------------|---------------------|----------------|----------------|----------------|----------------|
> | 0.0001                  | 80.5±0.9       | 80.3±2.8       | 79.8±1.9            | 79.6±1.2       | 79.4±1.0       | 80.6±1.7       | 80.1±1.7       |
> | 0.001                   | 80.8±1.2       | 80.7±1.8       | **81.5±2.0**        | 80.9±1.7       | 79.2±2.3       | 78.9±2.1       | 79.1±2.0       |
> | 0.01                    | 78.6±2.9       | 80.3±2.3       | 81.0±1.9            | 79.3±4.0       | 79.9±2.3       | 79.6±3.0       | 77.5±4.0       |
> | 0.1                     | 69.0±2.8       | 68.3±3.6       | 68.2±5.6            | 67.2±4.4       | 67.9±4.9       | 69.9±4.5       | 67.0±5.2       |
>
> **4. Segmentation results: Improvement on PointSegDA is smaller (≤2 mIoU). Provide qualitative visualizations or per-part IoU to show where CurvRec helps.**
>
> Thank you for your valuable comments. We present segmentation visualization results in Figure 4 (Appendix A.3, Page 24) for the PointSegDA dataset. Different segmentation parts are highlighted using different colors. We selected three domains—MA, MF, and SA—to demonstrate the effectiveness of our CDND method, which achieves state-of-the-art performance with a significant margin over the second-best method. In the figure, the last row labeled “GT” shows the ground truth. Our CDND results are shown in the third row. The “Unsupervised” row represents results without any domain adaptation technique. From Figure 4, it is evident that CDND produces segmentation results most closely aligned with the ground truth, outperforming both RS and the unsupervised baseline.  Additionally, It is worth noting that other methods also report small increases in mIoU margins on PointSegDA dataset.

---

> ### Author Response · Authors · 2025-07-02
> **Rebuttal**
>
> **Requested Changes:**
>
> **1. Equation for $I_e$ is unclear. The explanation for $I_e$ suggests “sum around both indices i and j,” but the equation instead shows a sum over off‐diagonal entries. Please rephrase to clearly state that  without the confusing “around both” wording.**
>
> We revised as requested. Please see Page 6 section 3.4. We mark the changes in red color.
>
> **2. Repeated symbols. Several symbols are reused in ways that create confusion. For instance, $K$ is first introduced as the number of eigenvalues, but then also appears as a Lipschitz constant. Likewise, C denotes a classifier, but later $\|C\|_F$ is used, which is ambiguous is this the classifier weight or its output matrix? Please use distinct symbols for these, such as K\lambda} for the eigenvalue count, and something like $\mathcal{C}$ or $P$ for the classifier output.**
>
> Thank you for your suggestions. We retain $K$ to denote the number of eigenvalues, and we now use $K_L$ to represent the Lipschitz constant to avoid ambiguity. The symbol $C$ refers to the classifier. The notation $\|C\|_F$ represents a distinct function: it applies the Frobenius norm after feeding the input through the classifier, rather than referring to the classifier itself. To further clarify, we now use $P_s$ and $P_t$ to denote the classifier outputs corresponding to the source and target samples, respectively.
>
> **3. Undefined distributions $\nu_s$ and $\nu_t$. These are introduced in Eq. (10), but never precisely defined. I can only guess that they are uniform empirical measures on batches, but this should be clearly stated in the text. Precise definitions are essential since the later theoretical results depend on the support of these measures.**
>
> Thank you for your suggestions. We add the definitions for them. Please see Page 8. We mark the changes in red color.
>
> **4. Motivation for combining D-NWD with PointMixup is not strong. D-NWD is used to align the domain distributions, while PointMixup is a data augmentation. Since both involve deformations, the paper should explain more explicitly why PointMixup alone is insufficient, and why an extra D-NWD term is necessary. At the moment, this feels ad hoc. An ablation or a stronger theoretical argument would improve confidence.**
>
> Thank you for your feedback. However, we would like to clarify that we do not combine D-NWD with PointMixup. We believe there may be a misunderstanding. In our ablation study (see Table 2 on page 10), the combination is between CurvRec and PointMixup (denoted as PCM), not D-NWD and PointMixup. We add CurvRec to PCM as part of an ablation study, in order to compare it with the previous method, DefRec+PCM.
> This ablative experiment is included to demonstrate that CurvRec + PCM outperforms DefRec + PCM, highlighting that our proposed CurvRec is a more advanced and effective deformation reconstruction method compared to DefRec.
>
> **5. Missing hyperparameter settings. These should be clearly listed, perhaps in an appendix.**
>
> Thank you for the reminder. All relevant hyperparameter settings are provided in Appendix A.2 under Implementation Details. Specifically, we set $k = 5$ for curvature region selection in deformation reconstruction. The number of center points used in D-NWD is equal to the number of regions, $N = 20$ for PointDA dataset and  $N = 40$ for PointSegDA dataset. For PointMixup (PCM), we adopt the same hyperparameters as used in the DefRec+PCM paper. The D-NWD loss weight corresponds to $\beta_1$, as listed in Tables 4 and 5. Additionally, the optimizer settings, including learning rate and the number of training epochs, are also detailed in Tables 4 and 5.

---

> > ### Comment · Reviewer_1AYa · 2025-08-04
> >
> > As for the distributions $\nu_s$ and $\nu_t$, their types remain unclear to me. Specifically, which kind of distribution (e.g., uniform, normal, or others) did you choose in your paper, and what is the rationale behind this choice?

---

> ### Author Response · Authors · 2025-07-03
> **Rebuttal**
>
> **Broader Impact Concerns:**
>
> **The authors do not provide a detailed broader impact statement. Since this paper focuses on domain adaptation for 3D point cloud data in tasks like classification and segmentation, its techniques could be deployed in safety‐critical systems (e.g., autonomous driving, robotics navigation) where failure in domain generalization may cause unintended consequences.**
>
> We acknowledge that our method has potential risks and can unintentionally propagate or exacerbate biases present in the source domain, particularly when transferring to real-world environments with unseen variations. Furthermore, although our method improves domain generalization, it does not guarantee safety in deployment, especially in high-stakes scenarios where failure has severe consequences. To mitigate such risks, we recommend:
>
> 1. Extensive safety evaluation before deployment in real-world applications.
>
> 2. Incorporation of uncertainty estimation mechanisms to detect out-of-distribution samples.
>
> 3. Periodic model audits and inclusion of domain experts during system integration.

---

> ### Author Response · Authors · 2025-08-04
> **Rebuttal**
>
> Thank you for your question. In our paper, $\nu_s$ and $\nu_t$ represent the probability measures of the source and target domains. These are general probability measures and are not limited to any specific type such as uniform or normal distributions. We intentionally do not assume a particular form for these distributions in order to keep our method and theoretical results as general and widely applicable as possible. In other words, our theoretical analysis works for any type of distribution. Restricting $\nu_s$ and $\nu_t$ to specific types would make the approach less flexible and reduce its usefulness in real-world domain adaptation settings, where the distributions are often unknown or complex.
>
> For a formal definition of probability measure, please refer to Appendix A.1, Definition 2.

---

> > ### Comment · Reviewer_1AYa · 2025-08-04
> >
> > Thank you for your response. In the methodology section, you state that the approach can generalize to any distribution measure; however, in the experiments, how do you perform numerical analysis when using arbitrary distributions?

---

> ### Author Response · Authors · 2025-08-04
> **Rebuttal**
>
> Thank you for your question. Following the standard theoretical framework for domain adaptation using optimal transport [1], any probability measure corresponds to an empirical measure in practice. Lemma 5 in Appendix of our paper shows the connection between a probability measure and its empirical measure. In Theorem 2 of our paper, we provide empirical measures of $\nu_s$ and $\nu_t$. Our Theorem 2 establishes the bound based on these empirical measures. In our experiments, we implemented D-NWD using Equation (10) from the main text, applied to the empirical measures.
>
> [1] I. Redko, A. Habrard, M. Sebban. Theoretical Analysis of Domain Adaptation with Optimal Transport, 2017.

---

> ### Author Response · Authors · 2025-08-08
> **Follow Up**
>
> Dear Reviewer,
>
> We are grateful for continual engagement during the discussion period. Given that we are approaching the discussion period deadline, we appreciate it if you let us know if there are any remaining concerns given our response. We are hopeful that with additional engagement, we can address all your concerns.
>
> Thank you,
>
> Our team

---

> > ### Comment · Reviewer_1AYa · 2025-08-27
> >
> > Dear authors,
> >
> > Thank you for your detailed response, which has addressed most of my concerns.
> >
> > I would strongly recommend incorporating these details into the experimental section. Furthermore, the paper would be greatly strengthened by an ablation study analyzing the results across different distributions.
> >
> > Best regards,

---

> ### Author Response · Authors · 2025-08-27
> **Rebuttal**
>
> Dear reviewer,
>
> We have already incorporated the requested clarifications into the revised version of the paper.
>
> We appreciate the suggestion regarding an ablation study on different distributions. However, it is important to note that the distributions of the source and target domains are determined by the datasets, and thus cannot be arbitrarily controlled or represented by standard distributions such as uniform or Gaussian in our experiments. We have conducted experiments on two widely used benchmark datasets, which together contain seven types of distributions (ModelNet, ShapeNet, ScanNet, ADOBE, FAUST, MIT, SCAPE). These distributions are naturally diverse and complex, and our results demonstrate that the proposed method remains effective across these varied settings. Therefore, while we cannot control or add distributions that we could test on, our experiments already provide strong evidence that the method generalizes well to different real-world domain distribution scenarios.
>
> Many thanks,
> Our team

---

### Review · Reviewer_q26F · 2025-07-03

**Summary Of Contributions:**

In this paper, the authors propose Curvature Diversity-Driven Nuclear-norm Wasserstein Domain Alignment (CDND), a novel UDA framework for point clouds. The contributions are twofold: (1) a practically effective method for modeling curvature variation, and (2) a theoretical lower bound on target-domain classification accuracy derived using the Wasserstein distance.

### Contributions

To be specific, their contributions can be listed below:

1. The method focuses deformation on flat (low-curvature) regions while preserving semantically informative (high-curvature) regions to maintain domain-invariant features.

2. Curvature variation is measured using entropy, allowing the identification and deformation of low-diversity regions.

3. A novel alignment strategy leverages both original and deformed features to enhance domain robustness through nuclear-norm Wasserstein distance.

4. A lower bound on target-domain classification accuracy is derived based on the NWD, providing theoretical insight into the alignment effectiveness.

### New Knowledge:

1. Connecting curvature diversity to deformation for domain invariance.

2. The Wasserstein alignment with deformed features reduces domain gap.

3. The combined deformation and WD in a trainable pipeline.

**Audience:**

Yes

**Claims And Evidence:**

Yes

**Requested Changes:**

1. Please include a table or plot showing the effect of different neighborhood sizes on accuracy and mIoU, as curvature estimation heavily depends on this parameter.

2. In Theorem 1, the assumption of equal probability measures for the two spaces lacks justification. Please either provide a proof that equal weighting minimizes the theoretical bound or include ablation results evaluating different weighting ratios.

3. Add a runtime analysis table comparing the computational overhead of the proposed method with baseline approaches to clarify its scalability and training cost.

4. Include a figure comparing original and deformed point clouds, with low-curvature regions highlighted to illustrate where and how the deformation is applied.

5. Validate the robustness of curvature estimation under real-world imperfections by introducing Gaussian noise to point coordinates and reporting the impact on performance.

**Strengths And Weaknesses:**

### Strengths:

1. The use of entropy-based curvature diversity for selective deformation is novel, enabling the preservation of semantically rich regions.

2. The paper provides both theoretical and empirical justification for targeting low-curvature regions for deformation, in contrast to prior random or volume-based methods.

3. It extends the Wasserstein distance to account for alignment between deformed and original features, and derives a theoretical bound on the target-domain classification error.

4. The paper is well-written, presents a clear workflow, and includes sufficient implementation details to support reproducibility.

### Weaknesses:

1. The analysis of curvature estimation is insufficient. In particular, ablation studies on parameters, such as the neighborhood size k and the number of principal components used in PCA, are missing. These parameters are important to the proposed method, as the accuracy of curvature estimation significantly influences domain adaptation performance.

2. The curvature estimation relies on clean local geometry, but the impact of real-world scanning artifacts (e.g., noise or occlusion) on its robustness is not discussed.

3. The joint optimization of curvature estimation, deformation generation, and Wasserstein distance alignment introduces considerable complexity compared to other point cloud UDA methods. The reported training time (~10 hours on an A100 GPU) may hinder scalability and practical deployment.

---

> ### Author Response · Authors · 2025-07-15
> **Rebuttal**
>
> **Weakness**
>
> **1. The analysis of curvature estimation is insufficient. In particular, ablation studies on parameters, such as the neighborhood size k and the number of principal components used in PCA, are missing. These parameters are important to the proposed method, as the accuracy of curvature estimation significantly influences domain adaptation performance.**
>
> Please see 1 in the Requested Changes.
>
> **2. The curvature estimation relies on clean local geometry, but the impact of real-world scanning artifacts (e.g., noise or occlusion) on its robustness is not discussed.**
>
> Please see 5 in the Requested Changes.
>
> **3. The joint optimization of curvature estimation, deformation generation, and Wasserstein distance alignment introduces considerable complexity compared to other point cloud UDA methods. The reported training time (~10 hours on an A100 GPU) may hinder scalability and practical deployment.**
>
> Please see 3 in the Requested Changes.

---

> ### Author Response · Authors · 2025-07-15
> **Rebuttal**
>
> **Requested Changes**
>
> **1. Please include a table or plot showing the effect of different neighborhood sizes on accuracy and mIoU, as curvature estimation heavily depends on this parameter.**
>
> Thank you for your valuable comments. We have added additional experiments to analyze the sensitivity of our method to hyperparameters involved in curvature estimation.
>
> **Neighborhood Size (k):**
> We tested the effect of varying the number of neighbors used in curvature computation. As shown in Table 9 (or below), the performance remains relatively stable (though the segmentation task is more sensitive) across different neighborhood sizes, suggesting that curvature estimation is not significantly sensitive to this parameter. Our chosen setting of 32 neighbors (denoted as CDND) achieves the best overall results for both classification (MS) and segmentation (FA) tasks.
>
> | Neighborhood  | 10             | 20         | 40         | CDND (32)      |
> | ------------- | -------------- | ---------- | ---------- | -------------- |
> | MS (Accuracy) | **84.1 ± 0.3** | 83.7 ± 0.4 | 83.6 ± 0.4 | **84.1 ± 0.3** |
> | FA (mIOU)     | 77.7 ± 2.9     | 76.7 ± 3.5 | 78.4 ± 2.6 | **81.5 ± 2.0** |
>
> **Principal Components:**
> Regarding the number of principal components, we clarify that we do not arbitrarily choose how many to use — the number is determined by the dimensionality of the input point cloud data. In our case, the input points lie in 3D space, so the covariance matrix is 3×3, yielding three eigenvalues and corresponding eigenvectors. All three principal components are used by definition, as shown in our implementation:
>
> ```covariance = np.cov(pc.T)  # 3x3 covariance matrix for 3D points
> covariance = np.cov(pc.T)  # 3x3 covariance matrix for 3D points
> w, v = np.linalg.eig(covariance)  # Eigenvalues and eigenvectors
> ```
>
> We use all three eigenvalues in computing curvature, and the normal vector is the eigenvector associated with the smallest eigenvalue, following standard practice for estimating local surface orientation and curvature in 3D geometry.
>
> **2. In Theorem 1, the assumption of equal probability measures for the two spaces lacks justification. Please either provide a proof that equal weighting minimizes the theoretical bound or include ablation results evaluating different weighting ratios.**
>
> Thank you for your insightful suggestion. In the theoretical section, we note that the results can be generalized to arbitrary sampling ratios by adjusting the mixture weights in the underlying probability measures. However, it is theoretically challenging to prove that a specific sampling ratio between original and deformed samples (e.g., 1:1, 2:3, etc.) is universally optimal. Instead, we provide experimental results to demonstrate that our chosen setting yields effective performance. We conduct experiments on the FAUST to ABODE task of the PointSegDA dataset. As shown in Table 8 (or below), we achieve the highest mIoU with the 1:1 and 4:3 ratios. This supports the intuition that a balanced and sufficiently diverse mix of original and deformed samples helps the model generalize more effectively across domains. It is also worth noting that our method is not highly sensitive to changes in the sampling ratio.
>
> | Ratio | 1:1        | 1:2        | 1:3        | 1:4        | 2:1        | 2:3        | 2:4        | 3:1        | 3:2        | 3:4        | 4:1        | 4:2        | 4:3            |
> | ----- | ---------- | ---------- | ---------- | ---------- | ---------- | ---------- | ---------- | ---------- | ---------- | ---------- | ---------- | ---------- | -------------- |
> | FA    | 81.5 ± 2.0 | 80.1 ± 1.1 | 79.3 ± 1.1 | 79.5 ± 3.9 | 79.5 ± 0.9 | 79.3 ± 1.1 | 79.5 ± 3.9 | 76.7 ± 2.8 | 79.9 ± 2.0 | 79.5 ± 3.9 | 77.2 ± 3.5 | 77.1 ± 3.0 | **81.7 ± 1.2** |

---

> ### Author Response · Authors · 2025-07-15
> **Rebuttal**
>
> **Requested Changes**
>
> **3. Add a runtime analysis table comparing the computational overhead of the proposed method with baseline approaches to clarify its scalability and training cost.**
>
> Thank you for your valuable comments. Table 11 (or below) presents the training time (in hours) for various baseline methods along with our proposed approach, CDND. As observed, our method does not introduce a significant computational overhead compared to the “Unsupervised” baseline (without any domain adaptation method), requiring extra 2.4 additional hours. This demonstrates a decent efficiency of CDND, especially when contrasted with methods like RS and GAST, which have significantly higher training costs. Furthermore, CDND’s training time is still comparable to DANN, PointDAN, and DefRec+PCM. Overall, CDND is practically viable without significantly sacrificing efficiency.
>
> | Method       | Time (hours) |
> | ------------ | ------------ |
> | Unsupervised | 8.0          |
> | DANN         | 9.0          |
> | PointDAN     | 9.2          |
> | RS           | 35.9         |
> | DefRec + PCM | 10.4         |
> | PCFEA        | 8.3          |
> | GAST         | 18.9         |
> | ImplicitPCDA | 9.8          |
> | CDND         | 10.4         |
>
> **4. Include a figure comparing original and deformed point clouds, with low-curvature regions highlighted to illustrate where and how the deformation is applied.**
>
> Thank you for your suggestions. We have figure for comparing original and deformed point clouds in the Figure 1, pipeline figure. In the Curvature Diversity-Based Deformation block, we have a lamp point cloud object as an example. The left-most lamp represents the original point cloud. The second lamp visualizes curvature values, where colors indicate **local curvature**: cooler colors correspond to lower curvature, and warmer colors represent higher curvature. The third lamp shows the curvature diversity computed over grouped regions (using Farthest Point Sampling (FPS) and *k*-NN). Here, the color scale reflects **curvature diversity**: cooler colors indicate lower diversity, while warmer colors correspond to higher diversity. Our method selects regions with the lowest curvature diversity (shown in blue) for deformation. Finally, the right-most lamp is the resulting deformed point cloud, where the selected regions have been replaced with random points.
>
> **5. Validate the robustness of curvature estimation under real-world imperfections by introducing Gaussian noise to point coordinates and reporting the impact on performance.**
>
> Thank you for your insightful comments. In Table 10 (or below), we introduce Gaussian noise to the point coordinates to evaluate the robustness of curvature estimation. For these experiments, the number of neighbors is fixed at 40. The classification task (ModelNet to ShapeNet) demonstrates robustness to noise, as the accuracy remains relatively consistent even with increasing noise variance. However, the segmentation task (FAUST to ADOBE) is more sensitive—its performance degrades significantly with higher noise levels. This increased sensitivity in segmentation arises because segmentation tasks require more precise, point-level geometric details to assign accurate labels to each point. Noise distorts these fine-grained features, making it harder for the model to distinguish between regions, especially in complex structures. In contrast, classification tasks benefit from more global shape features and can tolerate localized noise to a greater extent.
>
> | Gaussian Variance | 0.0001     | 0.01       | 0.1        | Clean          |
> | ----------------- | ---------- | ---------- | ---------- | -------------- |
> | MS (Accuracy)     | 83.2 ± 0.5 | 83.5 ± 0.4 | 82.7 ± 1.5 | **83.6 ± 0.4** |
> | FA (mIOU)         | 79.9 ± 1.3 | 75.8 ± 5.1 | 56.4 ± 1.9 | **78.4 ± 2.6** |

---

> ### Author Response · Authors · 2025-08-08
> **Follow Up**
>
> Dear Reviewer,
>
> We thank you for your time and feedback. Given that we are approaching the discussion period deadline, we appreciate it if you let us know if there are any remaining concerns given our response. We are hopeful that with additional engagement, we can address all your concerns.
>
> Thank you,
>
> Our team

---

> > ### Comment · Reviewer_q26F · 2025-08-27
> > **Post Rebuttal**
> >
> > Dear Authors,
> >
> > Thank you for the detailed and point-by-point responses. I have read through the response. The author have fully addressed all of my concerns. Thus, I would recommend acceptance.
> >
> > Best,
> >
> > Reviewer q26F

---

> > > ### Author Response · Authors · 2025-08-27
> > > **Follow-Up with Reviewer q26F**
> > >
> > > Dear Reviewer q26F,
> > >
> > > Thank you for reading our response. We are glad that we were able to address your concerns.
> > >
> > > Best regards,
> > >
> > > Our team

---

### Review · Reviewer_WVuD · 2025-07-30

**Summary Of Contributions:**

This paper proposes a novel method called the Curvature Diversity-Driven Nuclear-Norm Wasserstein Domain Alignment (CDND) for Unsupervised Domain Adaptation in 3D point cloud classification and segmentation. The main contributions are two-folds: (1) a deformation reconstruction task that selectively deforms regions with low curvature diversity, thereby encouraging the model to focus on extracting features from semantically rich regions; and (2) an extension of the Nuclear-norm Wasserstein Discrepancy (NWD) that aligns feature distributions of both original and deformed samples, enhancing domain alignment under geometric variations.

**Audience:**

Yes

**Claims And Evidence:**

Yes

**Requested Changes:**

1. Expand the introduction to clearly explain the motivation for the proposed modules. For example, provide justification for choosing entropy as the metric to quantify curvature diversity.
2. Although Section 3.2 briefly discusses the relationship between curvature and semantic richness, it is recommended to provide a more detailed explanation in the introduction.
3. Clarify and Support Empirical Findings: Provide additional theoretical analysis or visualizations to support the use of entropy over standard deviation in curvature measurement. Also, offer an explanation for the observation that CurvRec(En)-High performs worse than CurvRec(S)-High, as this could reveal limitations or sensitivities in the entropy-based region selection strategy.
4. Update Related Work: Include discussion of more recent point cloud UDA methods, such as SD and GLRV, in the Related Works section to improve the completeness and relevance of the literature review.
5. Self-Supervised Global-Local Structure Modeling for Point Cloud Domain Adaptation with Reliable Voted Pseudo Labels (GLRV, 2022)
Self-Distillation for Unsupervised 3D Domain Adaptation(SD, 2023)

**Strengths And Weaknesses:**

Strengths:
++ Curvature entropy as a proxy for semantic richness is intuitive and effective.
++ The paper provides solid mathematical analysis showing generalization bounds under D-NWD.
++ The ablation studies are well-executed and insightful, verifying the effectiveness of each component (CurvRec, D-NWD, SPST).

Weaknesses:
-- The writing clarity needs improvement. The introduction is too brief and lacks a thorough explanation of the motivations behind the proposed modules. For example, it is unclear why curvature is assumed to correlate with semantic richness, and why entropy is chosen as the metric for curvature diversity. These choices should be better justified to strengthen the intuition behind the method.
-- Although Table 2 shows that CurvRec(S)-Low and CurvRec differ in performance, indicating the effectiveness of using entropy to measure curvature diversity, this empirical finding lacks theoretical or visual support. Moreover, the authors do not explain why CurvRec(En)-High underperforms CurvRec(S)-High, which raises questions about the consistency and reliability of the entropy-based selection mechanism.
-- The Related Works section is missing discussion of some recent advances in UDA for point clouds, such as methods like SD and GLRV, which should be included for a more comprehensive and up-to-date review.

---

> ### Author Response · Authors · 2025-07-30
> **Rebuttal**
>
> **Weaknesses**
>
> See responses for Requested Changes.
>
> **Requested Changes**
>
> **1&2. Expand the introduction to clearly explain the motivation for the proposed modules. For example, provide justification for choosing entropy as the metric to quantify curvature diversity. Although Section 3.2 briefly discusses the relationship between curvature and semantic richness, it is recommended to provide a more detailed explanation in the introduction.**
>
> Thank you for pointing out the need to clarify the motivation behind our entropy-based curvature diversity metric. We have revised the introduction accordingly (with changes marked in red). Specifically, we add explanation for the relationship between curvature diversity and semantic richness and highlight that entropy effectively captures the variability of curvature within a region. This justifies our decision to deform low-entropy (less informative) regions while preserving high-entropy (semantically rich) ones.
>
>
> **3. Clarify and Support Empirical Findings: Provide additional theoretical analysis or visualizations to support the use of entropy over standard deviation in curvature measurement. Also, offer an explanation for the observation that CurvRec(En)-High performs worse than CurvRec(S)-High, as this could reveal limitations or sensitivities in the entropy-based region selection strategy.**
>
> We appreciate the reviewer’s suggestion to clarify the distinction between entropy and standard deviation in curvature-based region selection. We provided visual comparisons on ModelNet-10 dataset in Appendix A.3, Figure 5. In these visualizations, warmer colors indicate regions with higher semantic richness and curvature diversity. We observe that the entropy-based measure more effectively highlights structurally and semantically important regions, such as the joints between object components. For example, in the lamp, both the connection between the cap and the supporting rod, as well as the joint between the rod and the base, are clearly highlighted in red by entropy, indicating high curvature diversity. Similarly, in the figure, the interfaces between the tabletop and its supporting legs are also captured with warm colors, emphasizing their semantic importance. In contrast, the standard deviation measure fails to consistently highlight these joint regions. Furthermore, in the plant example, most of the leaves and flower structures—which are semantically rich and geometrically intricate—are highlighted by entropy with warm tones. However, these same regions are mostly shown in cooler colors (e.g., blue/green) under the standard deviation measure, indicating a failure to recognize their informative structure. Overall, these visualizations demonstrate that entropy provides a reliable and semantically aligned indicator of curvature diversity.
>
> Regarding the observation that CurvRec(En)-High performs worse than CurvRec(S)-High, we would like to clarify that this result actually supports the superiority of entropy in identifying semantically rich regions. Entropy is more effective than standard deviation in capturing areas with high curvature diversity and semantic importance. In the CurvRec(En)-High experiment, we intentionally deform the parts identified as having the highest curvature diversity—i.e., those that are semantically rich. Since entropy more accurately identifies these meaningful regions, deforming them results in a greater loss of critical geometric information, which negatively impacts model performance. On the other hand, CurvRec(S)-High, which uses standard deviation to select parts for deformation, may incorrectly classify geometrically simple or less informative regions as “important.” As a result, fewer truly semantically rich regions are deformed, and more informative structures are preserved. This allows the model to continue learning from these preserved rich regions, leading to better performance compared to CurvRec(En)-High. Therefore, the fact that CurvRec(En)-High leads to a larger performance drop actually reinforces our claim: entropy is more accurate in identifying semantically rich parts, and its deformation has a more significant impact on the model.
>
> All changes have been included in the revised version of the manuscript. They are highlighted in red in Appendix A.3.
>
> **4&5. Update Related Work: Include discussion of more recent point cloud UDA methods, such as SD and GLRV, in the Related Works section to improve the completeness and relevance of the literature review. Self-Supervised Global-Local Structure Modeling for Point Cloud Domain Adaptation with Reliable Voted Pseudo Labels (GLRV, 2022) Self-Distillation for Unsupervised 3D Domain Adaptation(SD, 2023).**
>
> Thank you for your valuable comments. We include these works in the new manuscript and mark the changes in red.

---

> > ### Comment · Reviewer_WVuD · 2025-09-08
> > **The response has addressed all my conern.**
> >
> > The response has addressed all my conern and I suggest such a manuscript be accepted.

---

> > > ### Author Response · Authors · 2025-09-08
> > > **Rebuttal**
> > >
> > > Dear Reviewer WVuD,
> > >
> > > Thank you again for your valuable comments and for reviewing our response! We are happy to address your concerns.
> > >
> > > Many thanks,
> > > Our team

---

> ### Author Response · Authors · 2025-08-08
> **Follow Up**
>
> Dear Reviewer,
>
> We thank you for your time and feedback. Given that we are approaching the discussion period deadline, we appreciate it if you let us know if there are any remaining concerns given our response. We are hopeful that with additional engagement, we can address all your concerns.
>
> Thank you,
>
> Our team

---

### Author Response · Authors · 2025-07-30
**Rebuttal is ready**

Dear Reviewers,

Thank you for your valuable comments and effort. The rebuttal is now ready. Please feel free to reach out if you have any questions. We look forward to further discussion with you!

Best,
Authors

---

### Decision · Action_Editor_dZ2W · 2025-09-16

**Recommendation:** Accept as is

**Audience:**

Yes

**Audience Explanation:**

Yes. The question of unsupervised domain adaptation is of interest to the community, and point cloud processing has an audience in the field.

**Claims And Evidence:**

Yes

**Claims Explanation:**

The three reviewers acknowledged the validity of the approach, its motivation, and the mathematical analysis done in the paper. They nonetheless expressed some concerns about the clarity of some aspects of the paper and some missing analyses of the method. The authors resolved all these concerns during the revision period, leading to all reviewers recommending acceptance.